# SparseMVC: Probing Cross-view Sparsity Variations for Multi-view Clustering

**Ruimeng Liu**[1], **Xin Zou**[2], **Chang Tang**[3]*
**Xiao Zheng**[4], **Xingchen Hu**[5], **Kun Sun**[1], **Xinwang Liu**[5]
[1]School of Computer Science, China University of Geosciences (Wuhan)
[2]The Hong Kong University of Science and Technology (Guangzhou)
[3]School of Software Engineering, Huazhong University of Science and Technology
[4]School of Computer Science, Hubei University of Technology
[5]College of Systems Engineering, National University of Defense Technology

## Abstract

Existing multi-view clustering methods employ various strategies to address data-level sparsity and view-level dynamic fusion. However, we identify a critical yet overlooked issue: *varying sparsity across views*. Cross-view sparsity variations lead to encoding discrepancies, heightening sample-level semantic heterogeneity and making view-level dynamic weighting inappropriate. To tackle these challenges, we propose Adaptive **Sparse** Autoencoders for **M**ulti-**V**iew **C**lustering (SparseMVC), a framework with three key modules. Initially, the sparse autoencoder probes the sparsity of each view and adaptively adjusts encoding formats via an entropy-matching loss term, mitigating cross-view inconsistencies. Subsequently, the correlation-informed sample reweighting module employs attention mechanisms to assign weights by capturing correlations between early-fused global and view-specific features, reducing encoding discrepancies and balancing contributions. Furthermore, the cross-view distribution alignment module aligns feature distributions during the late fusion stage, accommodating datasets with an arbitrary number of views. Extensive experiments demonstrate that SparseMVC achieves state-of-the-art clustering performance. Our framework advances the field by extending sparsity handling from the data-level to view-level and mitigating the adverse effects of encoding discrepancies through sample-level dynamic weighting. The source code is publicly available at https://github.com/cleste-pome/SparseMVC.

## 1 Introduction

Multi-view learning has emerged as a powerful paradigm for leveraging complementary information across multiple perspectives, significantly improving the performance of unsupervised learning tasks such as clustering [1, 2, 3, 4, 5]. At the same time, the sparsity of multi-view data has become a pivotal focus of research, with numerous studies proposing solutions from perspectives such as activation functions [6], tensor decomposition [7, 8], and variational autoencoders [9]. Nevertheless, while prior methods focus on designing advanced approaches to address data sparsity, they often overlook a fundamental aspect—the potential variations in sparsity across different views. Given that multi-view data consists of multiple views originating from distinct sources, and sparsity is a pervasive characteristic in multi-view data [10, 11, 12, 13, 14]. Building upon the factual observations, a natural question arises: "*Does there exist a phenomenon of varying sparsity across views?*"

---

*Corresponding author: Chang Tang (tangchang@hust.edu.cn).

39th Conference on Neural Information Processing Systems (NeurIPS 2025).

To quantify cross-view sparsity variations, we define the sparsity ratio $s_v$ for the $v$-th view:

$$s_v = \frac{1}{N \cdot F} \sum_{j=1}^{N} \sum_{i=1}^{F} \boldsymbol{I}[x_{i,j}^v = 0], \tag{1}$$

where $N$ refers to the number of samples, $F$ refers to the feature dimension, $x_{i,j}^v$ represents the $i$-th feature of the $j$-th sample in the $v$-th view, and the indicator function $\boldsymbol{I}$ takes the value of one if $x_{i,j}^v$ equals zero, and zero otherwise. Zero-valued features $x_{i,j}^v$ suggests missing dimensions or data collection errors. Our statistical and computational analysis addresses the question posed earlier and reveals that sparsity variations across views not only exist, but are widely prevalent in diverse multi-view data, as illustrated in Fig. 1.

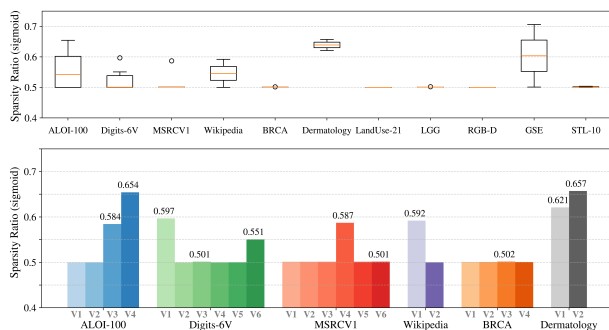

The disparity in view sparsity presents a multifaceted challenge: highly sparse views, lacking sufficient informative content, are prone to underfitting, whereas less sparse views, often burdened with redundant or irrelevant features, are susceptible to overfitting. Applying a uniform encoder architecture or regularization strategy across such heterogeneous views compromises representational consistency and limits the model's capacity to extract complementary cross-view information. To resolve this, we adopt an adaptive and sparsity-aware encoding strategy tailored to individual views. This requires rethinking the autoencoder design to accommodate view heterogeneity and structural disparities. Our solution is to design an autoencoder capable of adaptively adjusting its constraints based on the sparsity ratio of each view, allowing its encoding form to evolve accordingly.

Figure 1: Sparsity ratios across views in multi-view datasets. TOP BOX PLOT illustrates the sparsity ratio distribution, which shows the median (orange line), interquartile range (box), and any outliers (points outside the whiskers). BOTTOM BAR PLOT presents the sparsity ratios for each view within each dataset. To provide a more comprehensive situation of sparsity variations, additional datasets are included: Digits [15], LandUse-21 [16], RGB-D [17]. GSE [18] and STL-10 [19]. The cross-view sparsity ratios have been processed to improve visualization using **the sigmoid function**.

Due to varying sparsity across views for the same sample, encoders can introduce semantic and representational inconsistencies, affecting subsequent stages [20, 21]. As a result, it becomes essential to dynamically assess the contribution of each view based on the features extracted from different types of encoders. To address this, we design a correlation-informed reweighting module that assigns sample-wise weights based on the correlations between global and local features, thereby balancing view contributions and discrepancies. The early fusion strategy integrates multi-view data through feature concatenation. Since our reweighting module uses the global latent representation encoded from early fusion to guide the subsequent dynamic weighted early fusion of local representations, we chose an early fusion approach that preserves the original feature values as much as possible.

In late fusion, we introduce a cross-view distribution alignment module to align feature distributions across views, enabling robust integration and supporting datasets with arbitrary view numbers while balancing global consistency and view-specific diversity. These three core components together form the SparseMVC framework, addressing sparsity inconsistencies, encoding discrepancies, and semantic heterogeneity in a step-by-step and interdependent design.

Table 1: Performance comparison of different strategies under extreme and minimal sparsity variation.

| Datasets | Sparsity of Different Views | Accuracy | | NMI | |
| --- | --- | --- | --- | --- | --- |
| | | [Ours] | [2nd] | [Ours] | [2nd] |
| ALOI-100 | [0.0001, 0.0001, 0.3415, 0.6383] | 82.21 ↑4.49 | 77.72 | 92.65 ↑1.78 | 90.87 |
| MSRCV1 | [0.0049, 0.0048, 0.0048, 0.3478, 0.0051, 0.0048] | 97.14 ↑5.71 | 91.43 | 94.22 ↑6.32 | 87.90 |
| LGG | [0.0040, 0.0038, 0.0078, 0.0037] | 83.15 ↑0.38 | 82.77 | 54.62 ↑0.49 | 54.13 |
| Synthetic3d | [0.0017, 0.0017, 0.0017] | 98.33 ↑0.16 | 98.17 | 92.01 ↑0.74 | 91.27 |

As presented in Table 1, we evaluate our method across a range of datasets exhibiting extreme sparsity disparities. For instance, in ALOI-100, the sparsity ratio spans from as low as 0.0001 to as high as 0.6644, revealing a substantial imbalance in information density across views. To effectively

address such heterogeneity, our framework incorporates dynamically adaptive autoencoders that adjust both the encoding process and sparsity-aware regularization in accordance with each view's sparsity profile. Unlike existing methods, which generally overlook the impact of cross-view sparsity variation, our targeted design enables SparseMVC to achieve consistently superior performance under severe disparity conditions. Moreover, it maintains strong competitiveness even in datasets with near-uniform sparsity variations, such as LGG and Synthetic3D, demonstrating both its responsiveness to sparsity imbalance and its robustness in more homogeneous scenarios.

To the best of our knowledge, this is the first work to explicitly identify, analyze, and define the problem of cross-view sparsity variations in multi-view data, and to propose a dedicated framework SparseMVC that offers a targeted and principled solution.

## 2 Related Work

For multi-view fusion, early methods [22, 23] assumed equal importance of views, ignoring view heterogeneity. Hence, dynamic weighting approaches have emerged, with attention-based methods [24, 25] leading, alongside loss optimization [26, 27], kernel techniques [28, 29], and subspace methods [30]. However, uniform view-level weighting fails to address intra-view variability, highlighting the need for sample-level dynamic weighting. Trust-based methods [31, 32, 33, 34] excel in supervised scenarios. For multi-view clustering, sample-adaptive fusion [35] has been proposed using Laplacian matrix divergence. In contrast, our framework dynamically computes sample weights via correlation calculation without additional loss, while adopting a decoupled design with independent global and view-specific autoencoders. More details can be found in Appendix A.2.

Sparse representation effectively captures essential features in sparse data by enforcing sparsity constraints [36] but struggles to handle the non-linear structures common in multi-view datasets [37]. Autoencoders excel at learning non-linear latent features [38, 39], yet their lack of sparsity enforcement limits their adaptability to varying sparsity rates across views. Sparse autoencoders combine the strengths of both sparse representation and standard autoencoders by incorporating sparsity constraints into the hidden layers [40], which have become a research focus in multi-view learning [9, 41].

## 3 Method

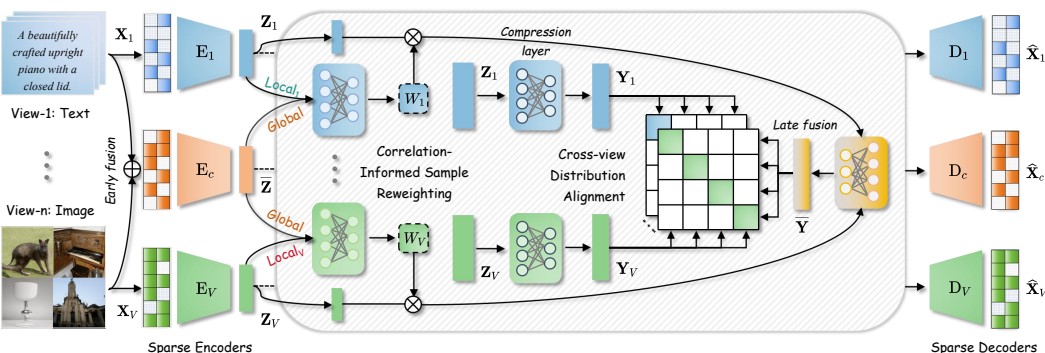

Figure 2: Overview of **SparseMVC**, a framework designed to address varying sparsity across views.

This section sequentially introduces the three key submodules of SparseMVC, as illustrated in Fig. 2. It begins by utilizing sparse autoencoders with adaptive constraints, which dynamically adjust the coding strategy based on the probed $s_v$, to generate latent features ($\boldsymbol{Z}$), making the reconstructed features ($\widehat{\boldsymbol{X}}$) approximate the original input features ($\boldsymbol{X}$). Subsequently, the correlation between the early-fused global features ($\bar{\boldsymbol{Z}}$) and view-specific features ($\{\boldsymbol{Z}_v\}_{v=1}^n$) guides the computation of sample-level weights ($\{\boldsymbol{W}_v\}_{v=1}^n$) via the attention mechanism within the correlation-informed sample reweighting module. Finally, the cross-view distribution alignment module enhances clustering performance by setting the late-fused global features $\overline{\boldsymbol{Y}}$ as the anchor latent representation, and then simultaneously aligning the multi-view feature distribution between $\overline{\boldsymbol{Y}}$ and each view-specific compressed feature ($\{\boldsymbol{Y}_v\}_{v=1}^n$). The algorithm of the framework can be found in Appendix A.1.

### 3.1 Sparse Autoencoder with Adaptive Constraints

To handle varying view sparsity rates, we propose the sparse autoencoder with adaptive constraints (SAA), extending traditional sparse autoencoders. SAA employs an adaptive loss function that integrates reconstruction and sparsity-aware entropy-matching as distinct constraints, wherein the adjustment is dynamically guided by view sparsity ratios formulated as prior knowledge.

The reconstruction loss, typically measured by mean squared error (MSE), quantifies the difference between the reconstructed output $\hat{\boldsymbol{x}}_j^v$ and the input $\boldsymbol{x}_j^v$ for the $j$-th sample:

$$\mathcal{L}_{\text{recon}}^v = \frac{1}{N} \sum_{j=1}^{N} \left( \hat{\boldsymbol{x}}_j^v - \boldsymbol{x}_j^v \right)^2, \tag{2}$$

where $N$ is the number of samples (batch size) for view $v$. Motivated by the widespread presence of sparsity variations in different views, our aim is to design a function that is positively correlated with $s_v$, allowing adaptive adjustments to both the encoder type and the strength of the sparsity constraints. We scale by $f(s_v)$ outside the loss rather than tuning $\rho$ in Eq. (4), please refer to Appendix B.2. The design of the adaptive weighting factor $f(s_v)$ follows a ReLU-like approach, adjusting the strength of $\mathcal{L}_{\text{entropy}}^v$ based on the probed input dataset sparsity $s_v$ for view $v$:

$$f(s_v) = \begin{cases} 0, & \text{if } s_v \leq \theta, \\ \frac{s_v - \theta}{1 - \theta}, & \text{if } s_v > \theta, \end{cases} \tag{3}$$

where the default value of $\theta$ is 0.01. Selecting the threshold $\theta$ for the sparsity ratio $s_v$ is based on the actual cross-view sparsity distribution of each dataset. For most datasets, the sparsity ratios exhibit a skewed distribution, with a significant concentration of both high and low values, as shown in Fig. 1 and Table 1. A $\theta$ value of 0.01 effectively captures these low-sparsity views. To enforce sparsity constraints, the entropy-matching loss is defined using Kullback-Leibler (KL) [42] divergence, encouraging the activations $\hat{h}_k^v$ in the hidden layer to align with a target sparsity level $\rho$. The entropy-matching loss and the sparsity loss derived from it are formulated as follows:

$$\mathcal{L}_{\text{sparse}}^v = f(s_v) \cdot \mathcal{L}_{\text{entropy}}^v = f(s_v) \cdot \sum_{k=1}^{H} \left( \rho \log \frac{\rho}{\hat{h}_k^v} + (1 - \rho) \log \frac{1 - \rho}{1 - \hat{h}_k^v} \right), \tag{4}$$

where $H$ refers to the number of units in the hidden layers of each sparse encoder, and $\rho$ is the target sparsity level. $\hat{h}_k^v$ represents the average activation of the $k$-th hidden unit for view $v$, which is clamped to lie strictly within the open interval from zero to one. Following [40], $\rho$ is set to 0.05, which is a well-validated choice that balances sparsity and the learning capacity of the autoencoder, allowing it to effectively capture key features in the data while avoiding overfitting to irrelevant features. The average activation is computed by:

$$\hat{h}_k^v = \frac{1}{N} \sum_{j=1}^{N} \sigma \left( \boldsymbol{W}_k^v \boldsymbol{x}_j^v + b_k^v \right), \tag{5}$$

where $\boldsymbol{W}_k^v$ is the weight matrix, $\boldsymbol{b}_k^v$ is the bias term for the $k$-th hidden unit in the $v$-th view, $\boldsymbol{x}_j^v$ is the input feature, and $\sigma(\cdot)$ denotes the ReLU activation function. When $s_v \leq \theta$, $\mathcal{L}_{\text{entropy}}^v$ is deactivated ($f(s_v) = 0$) and the sparse autoencoder degenerates into a standard autoencoder. Conversely, for input views where $s_v > \theta$, $f(s_v)$ exhibits a linear increase with $s_v$, ensuring that the sparsity constraint becomes more prominent for highly sparse inputs. Based on the above, the loss function for

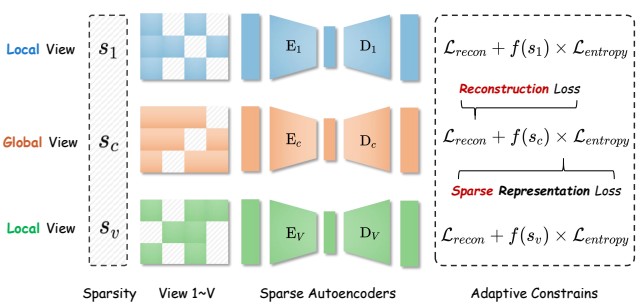

Figure 3: Sparse autoencoder with adaptive constraints.

SAA is then formulated as follows:

$$\mathcal{L}_{\text{SAA}} = \sum_{v=1}^{V} \left( \mathcal{L}_{recon}^{v} + \mathcal{L}_{\text{sparse}}^{v} \right). \tag{6}$$

As shown in Fig. 3, this piecewise linear design dynamically aligns the sparsity constraint with the input sparsity level and customizes the encoding strategy for each view.

## 3.2 Correlation-Informed Sample Reweighting

While SAA balances reconstruction and sparsity, it introduces sample-specific encoding inconsistencies due to differences in sparsity. Additionally, while networks adjust weights through layer updates, this weighting alone cannot address the lack of communication between autoencoders. On the above bases, we drew inspiration from the concept of multi-head attention and designed the correlation-informed sample reweighting module (CSR), which leverages correlations between the early-fused global features ($\bar{\boldsymbol{Z}}$) and the view-specific local features ($\boldsymbol{Z}_v$), then computes sample-specific weights. This cascading design is devised to achieve two objectives: mitigating the encoding inconsistencies introduced by SAA and leveraging globally fused features, which preserve the relatively high-fidelity patterns of the original data structure, to supervise the computation of view correlations.

CSR adopts a simplified structure inspired by multi-head attention, which captures similar effects while deviating from the standard formulation. Initially, CSR takes $\bar{\boldsymbol{Z}} \in \mathbb{R}^{N \times F}$ and $\boldsymbol{Z}_v \in \mathbb{R}^{N \times F}$ as input, and projects them into the query, key, and value spaces through parallel linear transformations:

$$\boldsymbol{Q} = \bar{\boldsymbol{Z}} \boldsymbol{W}_Q, \ \boldsymbol{K}_v = \boldsymbol{Z}_v \boldsymbol{W}_K, \ \boldsymbol{V}_v = \boldsymbol{Z}_v \boldsymbol{W}_V, \tag{7}$$

where $\boldsymbol{W}_Q, \boldsymbol{W}_K, \boldsymbol{W}_V \in \mathbb{R}^{F \times F}$ are the learnable weight matrices, generating the query matrix $\boldsymbol{Q} \in \mathbb{R}^{N \times F}$, which encapsulates global semantic information, and the key matrix $\boldsymbol{K}_v \in \mathbb{R}^{N \times F}$, which capture view-specific features for each view. Our primary objective is to quantify inter-view relationships by evaluating attention scores between queries and keys, without generating new feature representations, thus omitting the value matrix $\boldsymbol{V}_v$. To compute the correlation between $\bar{\boldsymbol{Z}}$ and $\boldsymbol{Z}_v$, we define the correlation score $\boldsymbol{C}_v \in \mathbb{R}^{N}$ for the $v$-th view based on Einstein summation convention as:

$$\boldsymbol{C}_v = \frac{\sum_{f=1}^{F} \boldsymbol{Q}_f \cdot (\boldsymbol{K}_f^v)^T}{\sqrt{F}}, \tag{8}$$

where $\sqrt{F}$ is the scaling factor equals to the square root of the dimension of the key vector. The correlation scores are normalized via the softmax function to produce sample-specific $\boldsymbol{W}_v \in \mathbb{R}^{N}$ as:

$$\boldsymbol{W}_v = \frac{\exp(\boldsymbol{C}_v)}{\sum_{v=1}^{V} \exp(\boldsymbol{C}_v)}, \tag{9}$$

that dynamically adjust the contribution of each corresponding sample in respective autoencoders.

## 3.3 Cross-view Distribution Alignment

Aligning features across views is a fundamental challenge in multi-view learning, as it is crucial for leveraging the complementary information provided by diverse views. The cross-view distribution Alignment module (CDA) addresses this issue by performing contrastive learning between the late-fused global features ($\overline{\boldsymbol{Y}}$) and the compressed features of individual views ($\boldsymbol{Y}_v$), ensuring effective alignment of multi-view features within a unified and shared latent space.

To mitigate the risk of dimensional collapse during alignment, potentially caused by an excessively large latent space, we introduce a compression layer before feeding the encoded view-specific features into the CDA. More details are provided in Appendix B.1. Specifically, $\boldsymbol{Y}_v \in \mathbb{R}^{N \times F}$ is obtained from $\boldsymbol{Z}_v$ via the compression layer. In parallel, the global features $\overline{\boldsymbol{Y}} \in \mathbb{R}^{N \times F}$ are as follows:

$$\overline{\boldsymbol{Y}} = \mathcal{F} \left( \sum_{v=1}^{V} \boldsymbol{W}_v \boldsymbol{Z}_v \right), \tag{10}$$

where fusion function $\mathcal{F}$ represents the late fusion layers and ensures that the transformed dimension matches $\boldsymbol{Y}_v$. The similarity matrix $\boldsymbol{S}_v$ between $\overline{\boldsymbol{Y}}$ and $\boldsymbol{Y}_v$ is defined as:

$$\boldsymbol{S}_v = \frac{\overline{\boldsymbol{Y}} \cdot (\boldsymbol{Y}_v)^T}{\tau}, \tag{11}$$

with $\tau$ denoting a temperature parameter that scales the similarity values. The sample pairs position indices in $\boldsymbol{S}_v$ are defined as $p$ and $q$. Positive pairs, which correspond to the same samples across views , are represented by the diagonal elements of $\boldsymbol{S}_v$, denoted as $\boldsymbol{S}_v^{p,p}$. Negative pairs, which involve different samples across views, are identified using a mask matrix $\boldsymbol{M}_v \in \mathbb{R}^{N \times N}$, where $\boldsymbol{M}_v^{p,q} = 1$ if $p \neq q$, and $\boldsymbol{M}_v^{p,q} = 0$ otherwise. The contrastive loss for each sample is:

$$\mathcal{L}_{con}^{p,v} = -\log\left(\frac{\exp(\boldsymbol{S}_v^{p,p})}{\sum_{q=1}^{N}\exp(\boldsymbol{S}_v^{p,q}) \cdot \boldsymbol{M}_v^{p,q}}\right), \tag{12}$$

where $\exp(S^{p,p})$ quantifies the similarity of positive pairs, and the denominator aggregates the exponential similarities of all pairs, weighted by the mask matrix $\boldsymbol{M}_v$. The overall CDA loss across all views is obtained by summing the individual losses for each view and averaging over all samples:

$$\mathcal{L}_{\text{CDA}} = \sum_{v=1}^{V} \frac{1}{N} \sum_{p=1}^{N} \mathcal{L}_{con}^{p,v}. \tag{13}$$

For contrastive learning, when samples of the same class are clustered in one view, the attraction exerted by positive pairs propagates to other views. In contrast, although the distinction between samples does not necessarily imply class disparity, repulsion among negative pairs enables view with greater discriminative power to transmit class separations to other views. This results in a mechanism that transforms both the alignment and misalignment information at the cross-view sample level into an objective, aiming to minimize intra-class while maximizing inter-class distances.

Regarding the role of CDA, the global view serves as an anchor, which is compared in parallel against each local view. To minimize the overall contrastive loss, the sample distribution of the global view is con-

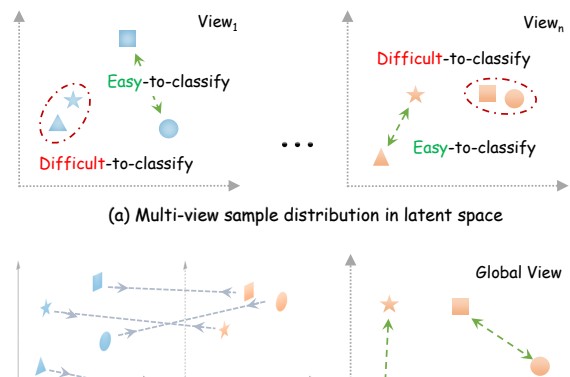

Figure 4: View distribution alignment based on contrast.

currently attracted toward all local views, thereby encouraging the features of each sample to converge more tightly in the latent space. From the perspective of the entire latent space, this process effectively facilitates overall distribution alignment. The rationale behind utilizing local views for distribution alignment with the global view to enhance class separability lies in the ability to leverage easily classified samples from one view to improve the distinguishability of harder-to-classify samples in another. The reasoning above, together with Fig. 4, illustrate the working principle of the CDA: utilizing contrastive learning as a tool, through the process of aligning the distribution of sample features across views, enhancing the differentiation of difficult-to-classify samples in one view by leveraging easy-to-classify samples in another, and ultimately achieving the goal of optimal alignment of features in the shared latent space.

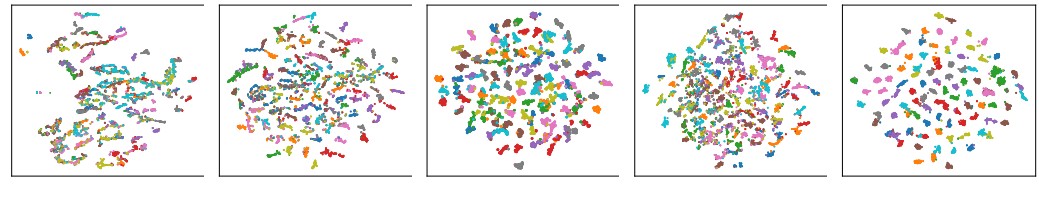

(a) GCFAgg (CVPR'23)   (b) CPSPAN(CVPR'23)   (c) SCMVC (TMM'24)   (d) MVCAN (CVPR'24)   (e) SparseMVC (Ours)

Figure 5: T-SNE visualization of the features learned with recently comparative methods (a-d) and ours (e) on the ALOI-100 dataset.

## 3.4 The Overall Loss Function of SparseMVC

The total loss $\mathcal{L}_{\text{total}}$ comprises the adaptive sparse autoencoder loss $\mathcal{L}_{\text{SAA}}$ in Eq. (6), preserving data fidelity and enforcing structured sparsity via $\mathcal{L}_{\text{recon}}$ and $\mathcal{L}_{\text{entropy}}$, and the cross-view alignment loss $\mathcal{L}_{\text{CDA}}$ in Eq. (13), ensuring consistent clustering across views:

$$\mathcal{L}_{\text{total}} = \sum_{v=1}^{V} \left( \mathcal{L}_{\text{recon}}^{v} + f(s_v) \cdot \mathcal{L}_{\text{entropy}}^{v} \right) + \lambda_{\text{CR}} \cdot \mathcal{L}_{\text{CDA}}, \tag{14}$$

where $\lambda_{\text{CR}}$ is the constraint ratio coefficient that controls the trade-off between $\mathcal{L}_{\text{SAA}}$ and $\mathcal{L}_{\text{CDA}}$.

## 4 Experiments

### 4.1 Experimental Settings

**Compared Methods.** Our proposed method is compared against the following 12 state-of-the-art multi-view clustering methods based on deep learning. DSMVC [43], COMPLETER [44], DCP [45], CVCL [46] and SCMVC [47] focus on dynamic contrastive learning; MFLVC [48], GCFAgg [3], DealMVC [49] combine feature fusion and consistency; DSMVC [43] and SDMVC [50] enhance consistency through discriminative learning; CPSPAN [51] and MVCAN [52] apply proxy supervision and prototype alignment.

**Benchmark Datasets.** The selected datasets span diverse domains: Image datasets include MSRCV1 [53] focusing on objects and scenes, Dermatology [54] on medical images, Out-Scene [55] on natural scenes, and ALOI-100 [56] on object recognition. Image-text datasets include Wikipedia [2], which provides website cross-modal data. Omics datasets include LGG [57] focusing on brain tumor genomics and BRCA [58] on breast cancer genomics. Synthetic3d [59] supports 3D object modeling and recognition. Detailed properties of datasets are listed in Table 2.

Table 2: Characteristics of kinds of multi-view datasets.

| Datasets | Samples | Clusters | Views | View Dimensions |
|---|---|---|---|---|
| **Images** | | | | |
| MSRCV1 | 210 | 7 | 6 | [1302, 48, 512, 100, 256, 210] |
| Dermatology | 358 | 6 | 2 | [12, 22] |
| Out-Scene | 2,688 | 8 | 4 | [512, 432, 256, 48] |
| ALOI-100 | 10,800 | 100 | 4 | [77, 13, 64, 125] |
| **Image-Text** | | | | |
| Wikipedia | 693 | 10 | 2 | [128, 10] |
| **Omics** | | | | |
| LGG | 267 | 3 | 4 | [2000, 2000, 333, 209] |
| BRCA | 398 | 4 | 4 | [2000, 2000, 278, 212] |
| **Synthetics** | | | | |
| Synthetic3d | 600 | 3 | 3 | [3, 3, 3] |

**Evaluation Metrics.** Accuracy (ACC) evaluates alignment with ground truth, normalized mutual information (NMI) measures shared information, purity (PUR) assesses cluster homogeneity. Adjusted Rand index (ARI), measuring clustering similarity, is partially utilized in experiments. For all metrics, higher values indicate better performance.

**Implementation Details** All experiments were conducted using Python 3.8.15 and PyTorch 1.13.1+cu116 on a Windows PC equipped with an AMD Ryzen 9 5900HX CPU, 32GB RAM, and an Nvidia RTX 3080 GPU (16GB). Models were trained using the Adam optimizer [60], a learning rate of 0.003, and a fixed seed of 50, with batch size equal to the dataset's sample count. Pre-training was performed uniformly for 300 epochs, while alignment training was conducted for 300 epochs for datasets with less than 2500 samples and 1000 epochs for larger datasets. For clustering, $k$-means [61] was applied with the number of clusters equal to the dataset categories and 100 initializations. During pre-training, global features $\mathbf{Z}_v$ derived from early fusion were used, while alignment training used late fusion features $\overline{\mathbf{Y}}$. Metrics were calculated as the average of 10 runs in the final epoch, with no fine-tuning performed for specific datasets. To ensure fairness, the hyperparameters for the comparison methods were determined based on either the default global settings or the configuration of the first dataset.

### 4.2 Comparative Results Analysis

Table 3 and 4 summarize the comparative results, leading to the following conclusions:

(1) Our method achieves state-of-the-art performance across eight diverse multi-view datasets, along with larger-scale datasets as shown in Table 8. These results validate the versatility of our approach

---

[2]https://dumps.wikimedia.org/

Table 3: Clustering results on small multi-view datasets. The top-ranked result is **bolded**, and the second-ranked result is underlined.

| Methods \ Datasets | Synthetic3d | | | LGG | | | Dermatology | | | BRCA | | |
|---|---|---|---|---|---|---|---|---|---|---|---|---|
| | ACC | NMI | PUR | ACC | NMI | PUR | ACC | NMI | PUR | ACC | NMI | PUR |
| COMPLETER [CVPR'21] [44] | 93.33 | 76.06 | 93.33 | 80.15 | 49.25 | 80.15 | 77.65 | 80.11 | 82.12 | 55.53 | 34.65 | 65.33 |
| DCP [TPAMI'22] [45] | 97.17 | 87.60 | 97.17 | 59.55 | 44.82 | 73.03 | 72.91 | 77.22 | 80.73 | 57.29 | 39.51 | 60.55 |
| MFLVC [CVPR'22] [48] | 90.67 | 72.59 | 90.67 | 79.03 | 49.73 | 79.03 | 58.10 | 56.20 | 62.85 | 55.53 | 27.74 | 60.05 |
| DSMVC [CVPR'22] [43] | 96.83 | 86.64 | 96.83 | 82.77 | 54.13 | 82.77 | 92.74 | 87.82 | 92.74 | 54.52 | 33.53 | 68.84 |
| SURE [TPAMI'22] [62] | 96.33 | 85.16 | 96.33 | 62.92 | 38.01 | 65.17 | 88.27 | 77.03 | 88.55 | 39.70 | 12.85 | 48.99 |
| DealMVC [MM'23] [49] | 87.50 | 72.07 | 87.50 | 72.28 | 40.55 | 72.28 | 45.53 | 31.13 | 45.53 | 59.55 | 32.79 | 61.56 |
| GCFAgg [CVPR'23] [3] | 96.67 | 85.54 | 96.67 | 55.06 | 22.95 | 61.80 | 88.27 | 79.25 | 88.27 | 51.51 | 32.41 | 61.31 |
| CPSPAN [CVPR'23] [51] | 97.83 | 90.15 | 97.83 | 63.30 | 30.53 | 63.30 | 76.26 | 84.63 | 85.20 | 66.83 | 34.48 | **74.12** |
| SDMVC [TKDE'23] [50] | 96.83 | 86.47 | 90.00 | 63.67 | 43.86 | 67.79 | 70.67 | 83.30 | 84.92 | 57.79 | 33.80 | 64.57 |
| CVCL [ICCV'23] [46] | 95.31 | 82.36 | 95.31 | 58.20 | 23.73 | 58.20 | 56.25 | 56.01 | 67.97 | 61.98 | 34.68 | 68.49 |
| MVCAN [CVPR'24] [52] | 98.17 | 91.27 | 94.59 | 59.55 | 42.57 | 27.18 | 58.38 | 66.73 | 51.58 | 57.79 | 35.70 | 32.24 |
| SCMVC [TMM'24] [47] | 97.00 | 87.11 | 97.00 | 73.41 | 39.76 | 73.41 | 93.85 | 88.44 | 93.85 | 50.25 | 30.70 | 60.80 |
| **SparseMVC (Ours)** | **98.33** | **92.01** | **98.33** | **83.15** | **54.62** | **83.15** | **95.25** | **89.86** | **95.25** | **70.10** | **44.90** | 70.85 |

and highlight its potential for a wide range of downstream tasks. In comparison, other approaches such as SCMVC, MVCAN, and CPSPAN achieved relatively good results on specific datasets but failed to maintain an advantage due to their limited generalizability across other datasets.

(2) Our method demonstrates stability in clustering performance, as evaluation metrics oscillate upward within a small range and stabilize with increasing epochs, showcasing robust results. In contrast, methods like SURE and DealMVC on ALOI-100 or CVCL and CPSPAN on Wikipedia fail to stabilize, with metrics either degrading significantly after peaking or fluctuating dramatically without consistent improvement. During the early stages of contrastive training when the embedding space's distribution remains uneven, potentially causing abrupt gradient fluctuations. To alleviate the instability, we adopt a dynamic fusion strategy in Sec. 3.2 and a pre-training approach in which only the autoencoder is trained initially.

(3) Although our model is specifically designed to address the challenge: across-view sparsity variations, it also achieves superior performance on dense datasets, such as BRCA, thereby demonstrating its adaptability and broad applicability. This is attributed to the inherent flexibility of our designed sparse autoencoder, which adaptively transitions into a conventional autoencoder when confronted with dense data, thereby prioritizing the reconstruction objective with greater emphasis.

(4) Compared to recent state-of-the-art methods, our approach demonstrates superior feature representation performance by producing clearer boundaries and more compact clusters, as shown in Fig. 5. Its most notable advantage is the ability to disentangle intra-class clusters while preserving inter-class separability, which leads to better scalability and robustness when handling data with large cross-view distribution disparities.

## 4.3 Convergence Analysis

By analyzing the training curve in Fig. 6, we observe the following key points: (1) The evaluation metrics generally exhibit an oscillatory increase followed by stabilization, with this stability being maintained as the number of training epochs progresses. This observation underscores the model's convergence and its robustness in maintaining stable clustering performance despite optimization challenges. (2) During the early stages of the alignment training phase, the evaluation metrics exhibit a brief dip, which is quickly followed by a recovery and subsequent stabilization at a higher level. The fluctuations observed in the evaluation metrics

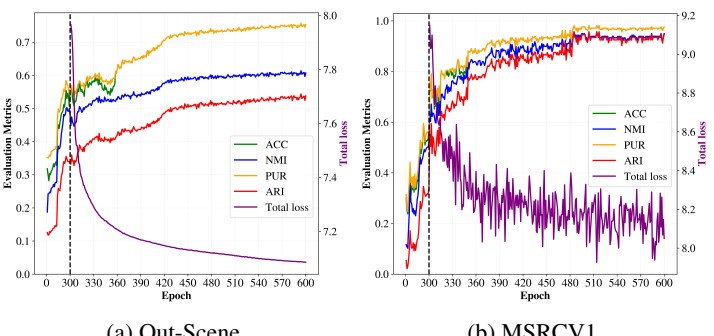

(a) Out-Scene        (b) MSRCV1

Figure 6: Convergence analysis of the training process. The left area of the vertical black dashed line represents the pre-training phase, while the right area stands for the view alignment training process.

can be attributed to $\mathcal{L}_{\text{CDA}}$ in Eq. (13), which necessitates the initialization of all network parameters except the autoencoder. (3) Despite significant fluctuations in the loss for small datasets, intriguingly, these fluctuations do not propagate to the evaluation metrics, suggesting that our approach effectively mitigates the influence of instability in the optimization algorithm on the clustering structure.

Table 4: Clustering results on big (Out-Scene & ALOI-100) and small multi-view datasets.

| Methods \ Datasets | Out-Scene | | | ALOI-100 | | | Wikipedia | | | MSRCV1 | | |
|---|---|---|---|---|---|---|---|---|---|---|---|---|
| | ACC | NMI | PUR | ACC | NMI | PUR | ACC | NMI | PUR | ACC | NMI | PUR |
| COMPLETER [CVPR'21] [44] | 69.79 | 55.39 | 69.79 | 30.70 | 62.12 | 33.63 | 57.14 | 53.10 | 59.31 | 90.00 | 87.90 | 90.00 |
| DCP [TPAMI'22] [45] | 56.03 | 45.59 | 56.32 | 34.01 | 60.28 | 37.32 | 45.31 | 43.16 | 46.32 | 25.71 | 23.25 | 27.14 |
| MFLVC [CVPR'22] [48] | 58.97 | 51.31 | 58.97 | 33.17 | 73.28 | 33.17 | 40.12 | 27.52 | 41.70 | 63.33 | 66.11 | 64.29 |
| DSMVC [CVPR'22] [43] | 62.13 | 53.01 | 64.25 | 71.52 | 90.87 | 72.72 | 60.32 | 54.74 | 62.19 | 64.29 | 54.29 | 64.29 |
| SURE [TPAMI'22] [62] | 60.97 | 48.09 | 60.97 | 10.13 | 34.19 | 11.90 | 50.65 | 39.97 | 54.11 | 91.43 | 85.84 | 91.43 |
| DealMVC [MM'23] [49] | 69.57 | 59.44 | 69.57 | 13.11 | 48.54 | 13.10 | 38.96 | 37.09 | 38.96 | 82.00 | 75.54 | 82.00 |
| GCFAgg [CVPR'23] [3] | 68.23 | 57.14 | 68.23 | 74.11 | 88.30 | 76.63 | 51.80 | 45.87 | 56.57 | 39.52 | 31.91 | 42.86 |
| CPSPAN [CVPR'23] [51] | 59.15 | 50.46 | 59.15 | 56.96 | 78.78 | 67.99 | 22.08 | 8.35 | 24.39 | 67.62 | 69.83 | 89.52 |
| SDMVC [TKDE'23] [50] | 56.03 | 46.18 | 59.93 | 52.02 | 74.70 | 56.56 | 55.99 | 53.98 | 62.05 | 59.52 | 52.51 | 45.24 |
| CVCL [ICCV'23] [46] | 73.51 | 59.59 | 73.51 | 21.86 | 43.13 | 23.29 | 14.17 | 42.81 | 32.69 | 48.44 | 84.57 | 90.62 |
| MVCAN [CVPR'24] [52] | 70.98 | 58.23 | 49.95 | 67.48 | 83.78 | 56.71 | 59.02 | 55.81 | 67.97 | 71.54 | 60.19 | 71.54 |
| SCMVC [TMM'24] [47] | 71.54 | 60.19 | 71.54 | 77.72 | 89.42 | 81.05 | 53.54 | 35.59 | 55.84 | 90.95 | 83.92 | 90.95 |
| **SparseMVC (Ours)** | **77.49** | **63.34** | **77.49** | **82.21** | **92.65** | **84.19** | **61.04** | 54.79 | **62.91** | **97.14** | **94.22** | **97.14** |

## 4.4 Ablation Study

**Loss Function** To ensure the rigor of ablation experiment, we selected three datasets with significant variations in view sparsity as shown in Fig. 1, ensuring the functionality of the SAA. We assessed the effectiveness of individual losses in the total loss Eq. (14) of SparseMVC, as presented in Table 5.

Specifically, we use $\mathcal{L}_{\text{recon}}$ as the baseline and find that adding either $\mathcal{L}_{\text{entropy}}$ or $\mathcal{L}_{\text{CDA}}$ improves performance. $\mathcal{L}_{\text{entropy}}$ yields substantial improvements on ALOI-100 and Dermatology, which have stronger variations in view sparsity. These improvements highlight the effectiveness of adaptive encoding and cross-view distribution alignment as robust constraints that contribute positively to the overall model training process. Moreover, when all losses are activated simultaneously, the model achieves optimal performance, suggesting that $\mathcal{L}_{\text{entropy}}$ and $\mathcal{L}_{\text{CDA}}$ complement each other synergistically. Notably, their integration does not introduce any mutual interference, further underlining the coherence and compatibility of these objectives in driving superior learning outcomes.

Table 5: Comparison of different loss function combinations.

| Datasets | Loss Function | | | Evaluation Metrics | | |
|---|---|---|---|---|---|---|
| | $\mathcal{L}_{\text{recon}}$ | $\mathcal{L}_{\text{entropy}}$ | $\mathcal{L}_{\text{CDA}}$ | ACC | NMI | PUR |
| ALOI-100 | ✓ | | | 45.27 | 71.21 | 30.41 |
| | ✓ | ✓ | | 66.35 | 81.65 | 70.62 |
| | ✓ | | ✓ | 64.11 | 80.23 | 67.33 |
| | ✓ | ✓ | ✓ | **82.21** | **92.65** | **84.19** |
| Dermatology | ✓ | | | 70.11 | 74.41 | 59.69 |
| | ✓ | ✓ | | 75.70 | 83.36 | 69.36 |
| | ✓ | | ✓ | 70.95 | 71.06 | 83.80 |
| | ✓ | ✓ | ✓ | **95.25** | **89.86** | **95.25** |
| MSRCV1 | ✓ | | | 58.57 | 48.63 | 32.76 |
| | ✓ | ✓ | | 70.48 | 65.27 | 52.08 |
| | ✓ | | ✓ | 92.38 | 87.62 | 92.38 |
| | ✓ | ✓ | ✓ | **97.14** | **94.22** | **97.14** |

**Components** Our approach focuses on view-level structural sparsity, specifically the sparsity variation across views within the same multi-view data. This differs from data-level sparsity methods, which typically apply uniform sparse encoding to all views without explicitly considering the heterogeneity of inter-view sparsity. To further validate the effectiveness of the proposed SAA module, we extend the ablation study in Table 5 by introducing two additional comparative settings: **(i)** *uniformly sparse encoding applied to all views, which mimics methods designed for data-level sparsity*; and **(ii)** *adaptive encoding tailored to each view*. On top of this, *we also ablate the CSR module*, which reweights the local features during the late fusion stage. Regardless of whether the CSR module is applied, the results in Table 6 show that the proposed SAA, which leverages adaptive autoencoders, remains effective and consistently achieves superior performance. Results further confirm that the effectiveness arises from the synergy between adaptive encoding and sample reweighting, rather than from the use of sparse autoencoders alone. Collectively, these findings confirm that our method is robust to varying sparsity across views and that the SAA and CSR modules function synergistically rather than independently.

Table 6: Ablation study on different components.

| Components \ Datasets | ALOI-100 | | | Dermatology | | | MSRCV1 | | |
|---|---|---|---|---|---|---|---|---|---|
| | ACC | NMI | PUR | ACC | NMI | PUR | ACC | NMI | PUR |
| all sparse autoencoders w/o CSR | 78.56 | 88.92 | 81.12 | 77.37 | 74.38 | 86.03 | 91.90 | 88.56 | 91.90 |
| all sparse autoencoders | 80.42 | 89.19 | 82.44 | 89.11 | 78.37 | 89.11 | 92.38 | 88.61 | 92.38 |
| adaptive autoencoders w/o CSR | 81.04 | 89.58 | 83.17 | 88.83 | 78.23 | 88.83 | 95.71 | 92.69 | 95.71 |
| adaptive autoencoders (Ours) | **82.21** | **92.65** | **84.19** | **95.25** | **89.86** | **95.25** | **97.14** | **94.22** | **97.14** |

## 4.5 Parameter Sensitivity Analysis

We selected the temperature parameter $\tau$ in $\mathcal{L}_{\text{CDA}}$ and the constraint ratio coefficient $\lambda_{\text{CR}}$, the ratio of $\mathcal{L}_{\text{SAA}}$ to $\mathcal{L}_{\text{CDA}}$, as the two parameters for analysis. Both coefficients were set to gradually increase from 0.1 to 1.9 with a step size of 0.3.

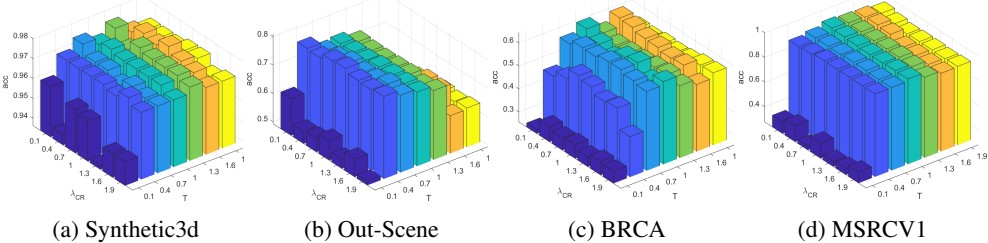

(a) Synthetic3d  (b) Out-Scene  (c) BRCA  (d) MSRCV1

Figure 7: The bar chart of clustering accuracy varying with different values of $\tau$ and $\lambda_{\text{CR}}$.

In Fig. 7, we can discover that the accuracy initially increases and then decreases as $\tau$ increases, remaining relatively stable within a range around 1.0. The influence of $\lambda_{\text{CR}}$ on clustering performance is comparatively minor, with a negligible impact when $\tau$ is within the range of 0.4 to 1.0. In light of Sec. 3.3, $\mathcal{L}_{\text{CDA}}$ incorporates the smoothing property of the logarithmic function, which diminishes the direct effect of $\lambda_{\text{CR}}$ adjustments on the gradient. In contrast, changes to $\tau$ significantly influence clusters separability and alignment performance by modulating the nonlinear response of the softmax function. A smaller $\tau$ enhances class separability, while a larger $\tau$ emphasizes global consistency. Therefore, we set the default values of $\tau$ and $\lambda_{\text{CR}}$ to 1.0 in the loss function.

Our method is largely insensitive to hyperparameter changes. To begin with, significant performance degradation only occurs when the temperature coefficient $\tau \leq 0.4$ or $\geq 1.6$. The extremely small value of 0.1 is chosen to probe the lower bound of performance degradation and is rarely used in practical applications. Furthermore, the performance fluctuation mainly occurs along the $\tau$-axis, whereas it remains relatively insensitive to changes in the constraint ratio coefficient $\lambda_{\text{CR}}$. In practice, it is the relative weighting between loss terms that is more commonly adjusted. In addition, the accuracy-axis was intentionally truncated to better highlight the differences, accentuating the visual disparity. Finally, regarding the concern that hyperparameters are not easy to tune in practice, our method maintains stable performance even under noticeable loss fluctuations, as illustrated in Fig. 6.

## 5 Conclusion

This paper highlights a frequently overlooked issue in deep multi-view learning: varying sparsity ratios across views. Therefore, we systematically define, quantify, and analyze cross-view sparsity variation as a fundamental characteristic of multi-view data. Our entire framework, SparseMVC, is designed to handle view-level sparsity variations with a complete data-driven and tightly integrated architecture. To tackle sparsity variation, we propose an adaptive encoding strategy that uses the sparsity ratio of each view as prior knowledge, enabling the encoder to switch between standard and sparse forms with appropriate constraint strengths. Additionally, we introduce a series of interdependent mechanisms to mitigate the side effects of representational divergence caused by non-uniform encoding. Specifically, a correlation-guided fusion strategy leverages global-to-local feature relationships from the early stages to guide the weighting of local features in late fusion. Moreover, a distribution alignment module structurally constrains the fused representations, enhancing cross-view complementarity in the final stage. Comprehensive experiments and detailed dissections of each module validate the efficacy of SparseMVC. We hope this work inspires greater attention to the intrinsic characteristics of data and to the design of architectures driven by data.

## Acknowledgements

This research was supported in part by the National Natural Science Foundation of China under grants 62522604 and 62476258, the Natural Science Foundation of Hubei Province under grant 2025AFA113, the Interdisciplinary Research Program of HUST under grant 2025JCY3024, and the Fundamental Research Funds for National Universities of China University of Geosciences (Wuhan) (No.2024XLB7).

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

## A    Algorithm and Comparison with Prior Work

### A.1    Algorithm

The training procedure for SparseMVC is described in Algorithm (1).

---

**Algorithm 1** Training Steps for SparseMVC

---

**Input:** Multi-view data $\{X_v\}_{v=1}^V$, cluster number $K$, and number of training epochs $E_{\text{pre}}$, $E_{\text{con}}$.
**Output:** Late-stage fusion representation $\overline{Y}$.
1: Initialize random seed and select Adam optimizer.
2: **for** $epoch = 1 : E_{\text{pre}} + E_{\text{con}}$ **do**
3:     Update $\{Z_v\}_{v=1}^V$ by minimizing $\{\mathcal{L}_{\text{recon}}^v\}_{v=1}^V$ and $\{\mathcal{L}_{\text{entropy}}^v\}_{v=1}^V$ utilizing Eqs. (2) and (4).
4:     Update $\bar{Z}$, formed by the concatenation of $\{Z_v\}_{v=1}^V$, utilizing Eq. (2) and Eq. (4).
5:     **if** $epoch > E_{\text{pre}}$ **then**
6:         Update weights $\{W_v\}_{v=1}^V$ by Eq. (9).
7:         Update $\overline{Y}$ by minimizing $\mathcal{L}_{\text{CDA}}$ utilizing Eq. (13).
8:     **end if**
9: **end for**
10: Perform $K$-means clustering on representation $\overline{Y}$.

---

### A.2    Comparison with Prior Work

To contextualize our contributions, we present a comparative discussion with representative dynamic weighting methods proposed in the literature. Dynamic weighting has been widely explored in multi-view learning, primarily through attention-based fusion mechanisms or optimization-driven strategies. Methods such as GCFAgg [3] employ attention to emphasize discrepancies among local views, aiming to refine feature alignment. Other approaches, including SPGMVC [63] and MAGCN [64], enhance feature encoding or perform view-level aggregation to improve representational quality. In parallel, techniques such as SCMVC [47], SCE-MVC [65] and TMC [66] utilize mutual information [67] or probabilistic priors to adaptively assign importance to views or losses. Furthermore, recent efforts have introduced view-invariant representations [46] and prototype-guided learning [52] to mitigate cross-view variability. Despite these advances, existing methods often overlook the inherent inconsistencies of sparsity between views, a phenomenon that can severely degrade the effectiveness of fusion [8, 9, 68]. In contrast, our proposed approach is motivated by the need to explicitly characterize and adapt to such cross-view sparsity variations. Specifically, our framework preserves global features obtained during early fusion and integrates both global and local view representations into the fusion process. The core of our design, the correlation-informed sample reweighting module, dynamically adjusts fusion weights based on the learned correlation between global and local views, thereby enabling fine-grained, sample-specific adaptation.

While MVASM [69] addresses the challenges of ambiguous class assignments in multi-view data by introducing an adaptive sparse membership matrix, our method introduces adaptive autoencoders with view-specific encoding strategies. Moreover, unlike existing techniques [70, 71, 72] that generally apply static or view-level weighting schemes, our method performs late-stage fusion reweighting conditioned on earlier global-local interactions. This design not only enhances fusion fidelity but also

elevates sparsity modeling from the feature level to the view level, offering a principled solution to a problem rarely addressed in the literature.

# B  Further Experiments

## B.1  Dimension and Feature Compression

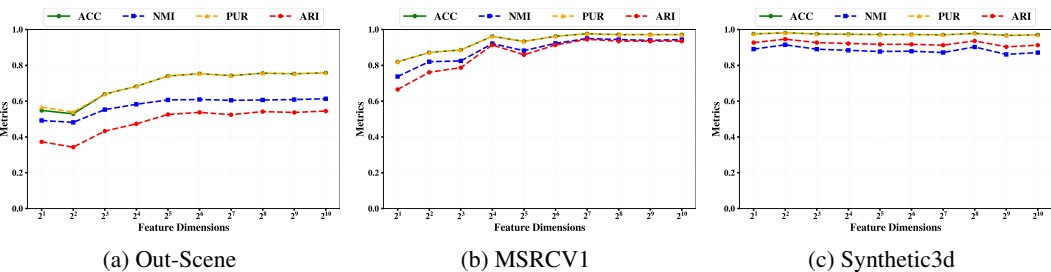

Figure 8: The clustering evaluation metric curves with respect to feature ($\boldsymbol{Z}_v$) dimension variations.

We tested the dimensions of view-specific features $\boldsymbol{Z}_v$ extracted by the autoencoder, varying from $2^1$ to $2^{10}$ in Fig 8. The experimental results demonstrated that as the feature dimension increased, all evaluation metrics (ACC, NMI, PUR and ARI) initially exhibited strong oscillations but progressively improved, eventually stabilizing within a small and well-performing range.

From the perspective of information-theoretic [73], low-dimensional features have limited encoding capacity that hinders capturing complex data patterns, whereas high-dimensional features offer greater capacity but risk dimensional collapse [74]. The feature compression layers that we designed compress and refine $\boldsymbol{Z}_v$ into $\boldsymbol{Y}_v$, mitigating the risks of overfitting and performance degradation often seen in contrastive learning. Accordingly, the model can avoid the negative effects of excessive dimensionality while leveraging its increased capacity for effective feature extraction. Ultimately, we selected 64 as the dimensionality of $\boldsymbol{Z}_v$ to balance computation and feature capacity.

## B.2  Selection of Scaling Factor

Table 7: Ablation study on scaling factor selection.

| The external scaling factor $f(s_v)$ | the internal ratio coefficient $\rho$ | MSRCV1 | ALOI-100 |
|---|---|---|---|
| | | **Mean accuracy** [Max accuracy] | |
| 0 or 1 | 1-$f(s_v)$ | **94.29** [95.71] | **77.75** [80.39] |
| $f(s_v)$ | 1-$f(s_v)$ | **95.24** [95.24] | **79.01** [80.08] |
| 0 or 1 | 0.05 | **95.71** [96.67] | **80.82** [81.51] |
| $f(s_v)$ | 0.05 | **97.14** [97.62] | **82.21** [82.93] |

To investigate the relative importance of the external scaling factor $f(s_v)$ and the internal ratio coefficient $\rho$ in modulating sparsity within the sparse autoencoder, we conducted a set of controlled experiments in which one parameter was kept constant while systematically varying the other. This decoupled analysis enables a clearer understanding of how each component contributes to the overall behavior of the model. Empirical results, as shown in Table 7, demonstrate that allowing $f(s_v)$ to be adaptively optimized while fixing $\rho$ at a reasonable constant leads to superior performance in multiple evaluation metrics. The smooth variation of the constraint strength is better than directly switching between zero and one, which represents whether the sparse constraint is applied. In contrast, varying $\rho$ while keeping $f(s_v)$ static yields relatively suboptimal results. These findings suggest that external view-level scaling plays a more critical role in capturing cross-view sparsity dynamics, highlighting the effectiveness of our adaptive design.

## B.3  The large-scale datasets

To evaluate the effectiveness and generalization ability of our proposed method, SparseMVC, we conduct additional experiments on large-scale datasets comprising over 8,000 samples, specifically GSE [18] and Animal [75]. The GSE dataset encompasses multi-omics data across 27 categories,

Table 8: Clustering results on the large-scale multi-view datasets.

| Datasets | GSE (8,200 samples) | | | Animal (11,673 samples) | | |
|---|---|---|---|---|---|---|
| View Sparsity Ratio | [0.877, 0.005] | | | [0.589, 0.179, 0.355, 0.467] | | |
| Methods | ACC | NMI | PUR | ACC | NMI | PUR |
| MVCAN [CVPR'24] | 71.40 | 75.41 | 72.29 | 12.06 | 10.25 | 15.48 |
| SCMVC [TMM'24] | 62.80 | 74.13 | 65.72 | 16.96 | 15.05 | 20.15 |
| SparseMVC (Ours) | **73.18** | **78.42** | **74.26** | **19.90** | **17.92** | **24.13** |

capturing a broad range of biological conditions, and is extensively utilized in bioinformatics research. Meanwhile, the Animal dataset consists of image data featuring animals with diverse attributes, spanning 20 categories. The detailed dataset information and comparison results, presented in Table 8, highlight the performance of our approach against the latest two methods employed in our manuscript. Clustering results on large-scale multi-view datasets further emphasize the superiority of our method, demonstrating its promising applicability in other data environments and scenarios.

## C  Visualization

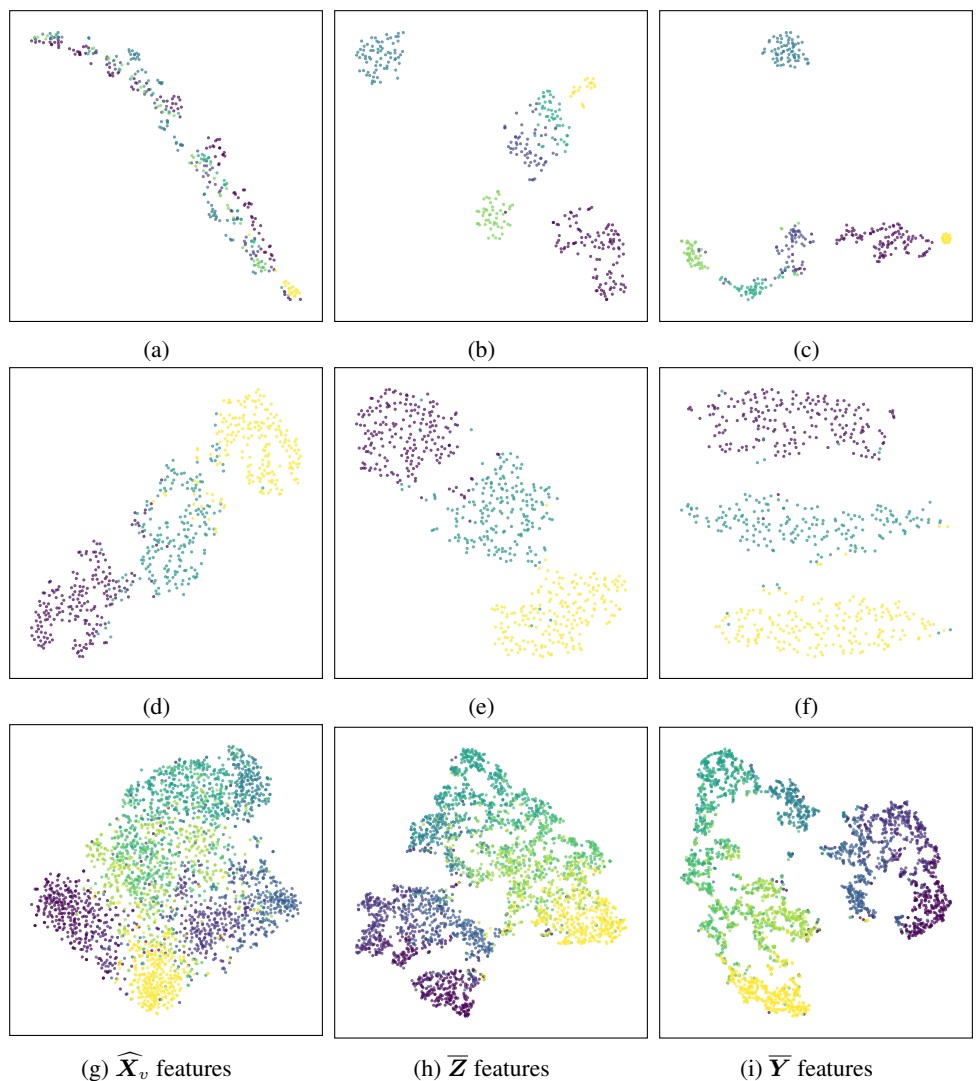

(a)

(b)

(c)

(d)

(e)

(f)

(g) $\widehat{X}_v$ features

(h) $\overline{Z}$ features

(i) $\overline{Y}$ features

Figure 9: The t-SNE visualizations of the original reconstruction features ($\widehat{X}_v$), the early fusion features ($\overline{Z}$), and the late fusion features ($\overline{Y}$) on datasets: Dermatology (a-c), Synthetic3d (d-f), and Out-Scene (g-i).

Figure 9 uses t-SNE for dimensionality reduction, mapping high-dimensional feature vectors into a two-dimensional space to facilitate a clearer and more insightful visualization of the distribution and structure of the data. The visualization reveals that the original reconstruction data, $\widehat{X}_v$, exhibit substantial interclass entanglement, with considerable overlap between different categories. In contrast, the preliminary global features, $Z_v$, derived from autoencoder pretraining, show a notable reduction in this entanglement, suggesting an early phase of disentanglement compared to the raw features. Further refinement through multi-view alignment results in the final fused features, $Y_{\text{global}}$, which exhibit the most pronounced disentanglement, yielding well-defined, separate clusters. These findings are further corroborated by the results in ablation analysis, which demonstrate that $Y_{\text{global}}$ consistently outperforms $Z_v$ in clustering tasks, particularly in distance-based methods like K-means. This performance enhancement underscores the pivotal role of the multi-view alignment and fusion process in improving feature separability and discriminative capacity, ultimately leading to more accurate and meaningful clustering results.

## D   Computational Complexity

The total loss function, as defined in Eq. (14), includes two main terms: $\mathcal{L}_{\text{SAA}}$ and $\mathcal{L}_{\text{CVDA}}$. The term $\mathcal{L}_{\text{SAA}}$ involves the reconstruction loss and the entropy-matching loss, both computed for each of the $V$ views, each containing $N$ samples. Since the feature dimension $F$ is constant and does not affect the computational scaling, the complexity of this term is determined by the number of views and samples, resulting in $O(VN)$. The term $\mathcal{L}_{\text{CVDA}}$ requires pairwise similarity computations between $N$ samples. For each pair of samples, the dot product computation has a constant cost of $O(1)$, leading to a total complexity of $O(N^2)$ for computing all pairwise similarities. The contrastive loss further requires computing the numerator, denominator, and logarithmic terms for each sample, which does not increase the overall scaling beyond $O(N^2)$. When combining both terms, the reconstruction and entropy-matching losses contribute $O(VN)$, while the pairwise similarity computations dominate with $O(N^2)$ when $N$ is sufficiently large. Thus, the overall computational complexity of the total loss function is $O(N^2)$.

## E   Theoretical Analysis

When learning from heterogeneous multi-view data, structural disparities, especially in sparsity patterns, present significant challenges to unified representation learning. Differences in modality, sampling granularity, and view incompleteness lead to varying information densities across views, making fixed encoder architectures and uniform loss formulations inherently suboptimal. To address this issue, we propose the Sparse Autoencoder with Adaptive Constraints (SAA), which dynamically adjusts the strength of sparsity constraints based on sparsity ratio $s_v$ of each view. This mechanism enables the encoder to balance compression and expressiveness in a view-aware manner, thereby facilitating the alignment of latent representations across structurally diverse views. In what follows, we provide a preliminary theoretical foundation for SAA, organized around a set of key questions that clarify its motivation, theoretical grounding, and coding formulation:

❶ **Why** should the sparsity constraint be adaptive?
When dealing with views of varying characteristics, uniform treatment of data may result in suboptimal representations, particularly when the model overly compresses informative in dense views or overemphasizes noisy in sparse ones. Accordingly, by the principle of minimal redundancy maximum relevance (mRMR) [76], we can establish trade-offs between different views, balancing the fidelity of reconstruction and the complexity of representation.

❷ **What** exactly are we coding or preserving?
Grounded in compressed sensing and optimally sparse representations [77, 78], we encode the essential structure of sparse inputs, which, though embedded in high-dimensional space, intrinsically conform to low-dimensional semantic subspaces. Therefore, sparse activation suffices to capture their core features, with sparse coding providing a stable, efficient, and redundancy-minimizing representation aligned with this inherent geometry.

## E.1  Why should sparsity constraints be adaptive?

**From Max-Dependency to mRMR and our view-wise coding.**  Let $S = \{x_i\}_{i=1}^m$ be a selected feature set and $c$ be the target. Peng et al. [76] formulate *max-dependency* as maximizing the mutual information between the *joint* selected features and the target:

$$\max_S \ D(S; c), \qquad D(S; c) = I(S; c) = I(x_1, \ldots, x_m; \ c). \tag{15}$$

Because directly estimating $I(S; c)$ is difficult when $m$ is not tiny, they introduce a first-order surrogate: *max-relevance* $D_{\mathrm{rel}}(S; c) = \frac{1}{|S|} \sum_{x_i \in S} I(x_i; c)$, and control redundancy by *min-redundancy* $R(S) = \frac{1}{|S|^2} \sum_{x_i, x_j \in S} I(x_i; x_j)$, then combine them as the mRMR criterion $\max\{D_{\mathrm{rel}}(S; c) - R(S)\}$ (difference form) [76]. In our setting, for each view $v$, the encoder produces a hidden code $\boldsymbol{h}_v = (h_1^v, \ldots, h_H^v)$ for input random variable $X_v$. Interpreting hidden units as the "selected features" $S$ and the input as the "target" $c$ (unsupervised but still information-theoretic), we use a weighted difference form to allow view-wise calibration, and the view-wise analogue of mRMR becomes:

$$\max \left[ \underbrace{\frac{1}{H} \sum_{k=1}^{H} I(h_k^v; X_v)}_{\text{max-relevance}} - \beta_v \underbrace{\frac{1}{H^2} \sum_{i=1}^{H} \sum_{j=1}^{H} I(h_i^v; h_j^v)}_{\text{min-redundancy}} \right], \tag{16}$$

where $\beta_v \geq 0$ is the redundancy/constraint strength for view $v$.

**Why $\beta_v$ depends on view sparsity $s_v$: redundancy scales with the effective degrees of freedom.**
SparseMVC explicitly measures the sparsity ratio $s_v$ of view $v$ as a prior statistic in Eq. (1). A larger $s_v$ indicates a higher fraction of zero (or missing/default) entries and thus a lower effective information density in $X^v$. In this regime, the first term in Eq. (16) maximizing relevance alone can encourage multiple hidden units to focus on the same limited set of informative patterns, which increases dependence among $\{h_k^v\}$ and inflates the redundancy term.

To relate redundancy to the sparsity regularizer used by sparse autoencoders, we follow the standard sparse autoencoder interpretation [40] and treat each hidden unit as a Bernoulli activation variable when discussing entropic quantities. Then, for any pair $(i, j)$,

$$I(h_i^v; h_j^v) = H(h_i^v) - H(h_i^v \mid h_j^v) \leq H(h_i^v), \tag{17}$$

where $H(\cdot)$ denotes Shannon entropy of the corresponding Bernoulli variable. This inequality shows that suppressing the entropies (and co-activations) of hidden units controls an upper bound on pairwise dependence, and thus provides a principled route to reducing redundancy when the view is highly sparse.

**SAA realizes adaptive mRMR via entropy-matching.**  SparseMVC instantiates the adaptive redundancy coefficient as a deterministic function of $s_v$ in Eq. (3) and implements redundancy control via the entropy-matching KL penalty (sparse autoencoder prior) on the mean activation $\hat{h}_k^v$ in Eq. (4). For numerical stability in KL computation, the estimated mean activation is clamped to remain strictly within $(0, 1)$ in implementation. This design matches the mRMR logic at the view level: $s_v$-large views are precisely those where redundancy inflation is most likely, so $f(s_v)$ increases the effective redundancy penalty; $s_v$-small views receive weak or no sparsity constraint (e.g., $f(s_v) = 0$ below $\theta$), preventing unnecessary relevance loss.

**Conclusion.**  According to the mRMR principle [76], effective representations should maximize relevance while minimizing redundancy. SparseMVC operationalizes this principle with a view-dependent redundancy strength, using the sparsity ratio $s_v$ to adaptively scale the KL-based sparsity penalty. As a result, sparse views benefit from stronger redundancy suppression, whereas dense views retain greater expressive capacity, yielding a principled, data-driven trade-off between information preservation and redundancy reduction across heterogeneous views.

## E.2  What exactly are we coding or preserving?

**This question concerns the rationale and effectiveness of using sparsity constraints and sparse autoencoders to represent sparse data.** Sparse data are characterized by low information density:

only a small fraction of entries are nonzero and carry informative structure, while many coordinates may correspond to missing/default states. This suggests that an effective representation should focus capacity on the informative components, rather than distributing it uniformly across all dimensions. Consequently, representations of sparse inputs are often encouraged to be sparse as well, activating only a small subset of latent units that capture the dominant patterns supported by the nonzero entries, thereby reducing redundancy.

**Sparse inputs and informative support.** Formally, let the $j$-th sample from view $v$ be denoted by $\boldsymbol{x}_j^v \in \mathbb{R}^{n_v}$, with sparsity level $\|\boldsymbol{x}_j^v\|_0 = k \ll n_v$. Define the (sample-dependent) support set $\Omega_j^v = \{i \mid x_{j,i}^v \neq 0\}$. In sparse views where zeros primarily represent missing/default entries, the informative variation is mainly concentrated on the restricted subvector $x_{j,\Omega_j^v}^v$, while the complement $x_{j,(\Omega_j^v)_c}^v$ contributes little signal. Thus, allocating excessive modeling capacity to the zero-valued region $(\Omega_j^v)_c$ may introduce unnecessary degrees of freedom and increase redundancy.

**Reconstruction error and support-aware desideratum.** Consider a linear decoder or dictionary $D_v \in \mathbb{R}^{n_v \times m_v}$ and a latent code $\alpha_j^v \in \mathbb{R}^{m_v}$. Let $D_{\Omega_j^v}^v$ denote the *row* submatrix of $D^v$ indexed by $\Omega_j^v$, and let $D_{(\Omega_j^v)_c}^v$ denote the *row* submatrix indexed by the complement of $\Omega_j^v$. Then the reconstruction error decomposes as:

$$\|\boldsymbol{x}_j^v - D_v \alpha_j^v\|_2^2 = \|x_{j,\Omega_j^v}^v - D_{\Omega_j^v}^v \alpha_j^v\|_2^2 + \|D_{(\Omega_j^v)_c}^v \alpha_j^v\|_2^2. \tag{18}$$

The second term measures spurious energy produced on coordinates that are zero in the input. A natural desideratum for sparse-view reconstruction is therefore to keep $\|D_{(\Omega_j^v)_c}^v \alpha_j^v\|_2^2$ small. Importantly, sparsity of $\alpha_j^v$ alone does not guarantee this term is exactly zero; rather, sparsity regularization reduces the degrees of freedom available to fit $(\Omega_j^v)^c$ and encourages the decoder to rely on a small number of atoms, which empirically helps suppress spurious energy outside the informative support.

**Sparse coding and recoverable representations.** Under this support-aware viewpoint, sparse coding seeks a representation of the form:

$$\alpha_j^{v*} = \arg\min_{\alpha} \|x_{j,\Omega_j^v}^v - D_{\Omega_j^v}^v \alpha\|_2^2 \quad \text{s.t.} \quad \|\alpha\|_0 \leq s, \tag{19}$$

with $s \ll m^v$. Classical results in sparse approximation characterize uniqueness in the noiseless exact-representation case, e.g., if $x_{j,\Omega_j^v}^v = D_{\Omega_j^v}^v \alpha$ and $s < \frac{1}{2}\text{spark}(D_{\Omega_j^v}^v)$, then the sparsest solution is unique [77]. Compressed sensing theory further establishes stable recovery guarantees under Restricted Isometry Property (RIP)-type conditions for standard $\ell_1$-based decoders [78]. These guarantees justify sparse coding as a canonical representational prior for signals whose structure is concentrated on a low-dimensional support.

**Sparse autoencoder as amortized sparse recovery.** A sparse autoencoder implements the above sparse recovery principle in an amortized and differentiable manner. For view $v$, the encoder produces hidden activations whose empirical mean is given by:

$$\hat{h}_k^v = \frac{1}{N}\sum_{j=1}^N \sigma(\boldsymbol{W}_k^v \boldsymbol{x}_j^v + b_k^v) = \frac{1}{N}\sum_{j=1}^N \sigma(\sum_{i=1}^F w_{i,k}^v x_{i,j}^v + b_k^v), \tag{20}$$

where $\sigma(\cdot)$ is chosen so that the quantity used in the KL term can be interpreted as a probability, and $\hat{h}_k^v$ is clamped to lie strictly within $(0,1)$ in implementation before computing the KL divergence. Enforcing $\hat{h}_k^v$ to match a small target sparsity level via a KL-divergence sparsity penalty imposes an $\ell_0$-like constraint in expectation on the latent representation. As a result, for each sparse input $\boldsymbol{x}_j^v$, only a small subset of hidden units tends to be active, which qualitatively mimics sparse coding behavior without explicitly solving a combinatorial optimization problem. Eq. (20) also makes explicit that sparsity is enforced through an empirical average (over samples or minibatches) rather than per-sample hard sparsity constraints.

**Conclusion.** Sparse coding theory and compressed sensing results [77, 78] provide a principled rationale for representing sparse-view signals with sparse codes under standard recoverability assumptions. A sparse autoencoder operationalizes this idea by combining reconstruction with a KL-based sparsity penalty, yielding compact representations that suppress redundant co-activations and emphasize patterns supported by the informative entries of the input.

# F Limitations and Future Work

While effective in addressing cross-view sparsity variations, the current approach has several limitations that suggest directions for further improvement. To begin with, SparseMVC demonstrates strong performance across a variety of multiview data, but its effectiveness in real-world applications with noisy or incomplete scenarios remains to be fully explored. Additionally, the use of contrastive learning inherently introduces computational overhead, making it unlikely to rank among the fastest available approaches. Moreover, the framework assumes well-aligned views, whereas slight inter-view misalignment may occur in real-world scenarios.

A potential direction for future work is to incorporate structural information across views, which can be modeled through priors such as graph connectivity, inter-view relational graphs, or mutual information constraints—techniques that have been extensively explored in prior multi-view representation learning research. In contrast, the present study primarily focuses on feature-level sparsity and its adaptive regularization, rather than modeling explicit structural dependencies. Incorporating structural priors may further enhance representation quality, particularly in scenarios where inter-view relationships are semantically meaningful.

