# OpenReview forum: "SparseMVC: Probing Cross-view Sparsity Variations for Multi-view Clustering"
_NeurIPS.cc/2025/Conference — NeurIPS 2025 spotlight_

### Official Review · Reviewer_h93U · 2025-06-18

**Clarity:** 3
**Significance:** 3
**Originality:** 3
**Rating:** 5
**Confidence:** 4

**Summary:**

This manuscript explicitly introduces and analyzes the phenomenon of cross-view sparsity variations in multi-view data, a challenge that has been largely overlooked by existing clustering methods. In response to this newly identified issue, the authors propose SparseMVC as a targeted and systematic solution. SparseMVC employs an adaptive sparse autoencoder that dynamically adjusts encoding constraints for each view according to its sparsity level, reweights sample contributions based on correlations between local and global features to mitigate inconsistencies, and aligns feature distributions across views to enhance clustering robustness and thereby improve generalization.

**Questions:**

(1) How does the correlation-informed sample reweighting module differ from attention-based or other dynamic weighting methods in multi-view clustering? What limitations of these prior approaches does it address?
(2) Why does the SparseMVC framework apply an additional transformation to the encoded latent features (Z), producing compressed representations (Y)? What motivates this design choice?
(3) What criteria guided the selection of the sparsity threshold (τ)?

**Ethical Concerns:**

["NO or VERY MINOR ethics concerns only"]

**Final Justification:**

Thanks for the author rebuttal. I am satisfied with the author response and would like to accept this work.

**Limitations:**

Yes

**Paper Formatting Concerns:**

From a comprehensive review of the manuscript, there are no major formatting issues apparent based on the provided content.

**Quality:**

3

**Strengths And Weaknesses:**

Strengths: The manuscript introduces a novel, data-driven approach to addressing cross-view sparsity variations, a challenge that has been largely overlooked in prior research. By focusing on sparsity at the view level, the authors offer a refined and effective solution, SparseMVC, which enhances both the accuracy and robustness of clustering results. SparseMVC is meticulously designed, incorporating three key components that collectively tackle the problem from multiple angles. Each module is distinct yet seamlessly integrated, ensuring a comprehensive approach to the complexities posed by varying sparsity across views. Furthermore, the paper is exceptionally clear and well-structured, beginning with a concise problem statement and thoroughly explaining the rationale behind the proposed solution, making the complex ideas both accessible and academically rigorous. Lastly, the significance of this work lies in its emphasis on addressing inherent data characteristics, rather than merely refining existing models. By shifting the focus toward data-driven model design, this paper opens up new avenues for future research in multi-view learning.

Weaknesses: While the method demonstrates strong performance across a variety of datasets, its effectiveness in real-world applications with noisy or incomplete data remains to be fully explored. Further analysis is needed to evaluate how well SparseMVC handles extreme cases of sparsity or data imperfections in practical scenarios. Additionally, while the model shows strong performance in the downstream clustering task, its applicability to other tasks remains unexplored. The figures, especially the diagrams illustrating the modules, could be improved, particularly in terms of the consistency of symbols and the definitions of certain input variables.

---

> ### Author Rebuttal · Authors · 2025-07-29
>
> Thanks for your acknowledgment of our method. We will address your questions one by one.
>
> ------
>
> > ***Question 1**:* How does **the correlation-informed sample reweighting module differ from attention-based or other dynamic weighting methods** in multi-view clustering? What **limitations of these prior approaches** does it address?
>
> Thanks. The key differences between our correlation-informed sample reweighting (CSR) module and existing dynamic weighting methods can be analyzed from two perspectives: design foundation and intended purpose.
>
> (1) For the foundation: Our method preserves the global features extracted during early fusion and incorporates both global and local features as inputs. Unlike attention-based fusion methods such as GCFAgg (CVPR'23) that emphasize disparities among local views, our CSR module performs dynamic view weighting based on the correlation between global and local features. It also differs from other strategies that depend on view-invariant representations (e.g., CVCL, ICCV'23), loss-based optimization (e.g., SCMVC, TMM'24), or prototype-driven fusion (e.g., MVCAN, CVPR'24).
>
> (2) For the purpose: Prior attention-based fusion methods aim to enhance feature encoding (e.g., SPGMVC, TKDE'21) or view-level aggregation (e.g., MAGCN, AIJ'22), and others like SCMVC (TMM'24) and TMC (TPAMI'22) emphasize view or loss weighting via mutual information or Dirichlet priors. In contrast, our method applies sample-specific weighting in the late fusion stage, guided by the correlation between global and local views established during the early fusion phase.
>
> Addressed limitations: The limitations of existing methods stem from the absence of an autoencoder capable of dynamically adjusting based on the sparse differences between views, as well as the inability to effectively leverage the relationship between globally fused features and local view features from earlier fusion stages. The SAA and CSR modules are specifically designed to address these two key shortcomings.
>
> ------
>
> > ***Question 2**:* Why does the SparseMVC framework **apply an additional transformation to the encoded latent features** $Z$, producing compressed representations $Y$? What **motivates** this design choice?
>
> Thanks. As shown in **Table 8 (Appendix B.1)**, larger latent feature dimensions in the autoencoder tend to improve overall performance. At the same time, excessively high-dimensional representations in contrastive learning can lead to dimensional collapse and unstable clustering. To address this trade-off, the compression layer applies dimensionality reduction to balance the performance of the CSR and CDA modules.
>
> ------
>
> > ***Question 3**:* What criteria guided **the selection of the sparsity threshold**?
>
> Thanks. Selecting the threshold value for the sparsity ratio $s_v$ is based on the actual cross-view sparsity distribution of each dataset. For most datasets, the sparsity ratios exhibit a skewed distribution, with a significant concentration of both high and low values, as shown in **Table 1**. A sparsity threshold value of 0.01 effectively captures these low-sparsity views.

---

> > ### Comment · Reviewer_h93U · 2025-08-06
> >
> > Thanks for the author rebuttal. I am satisfied with the author response and would like to accept this work.

---

> > > ### Author Response · Authors · 2025-08-06
> > >
> > > Dear Reviewer h93U,
> > >
> > > We are truly thankful for your reply. Your insights have greatly contributed to refining our manuscript, and we sincerely appreciate your continued support during this revision process.
> > >
> > > If you have any additional recommendations, we would be glad to take them into consideration. Once again, thank you for your time and kind attention.
> > >
> > > Best wishes,
> > >
> > > Authors

---

### Official Review · Reviewer_bBMF · 2025-06-24

**Clarity:** 3
**Significance:** 4
**Originality:** 3
**Rating:** 5
**Confidence:** 4

**Summary:**

The paper rigorously quantifies and identifies variations in view-level sparsity ratios across a wide range of multi-view data, presenting an innovative adaptive, unsupervised representation learning network. While prior research on sparsity in multi-view clustering has predominantly focused on addressing sparsity at the dataset level, this work makes a significant contribution by targeting internal view sparsity discrepancies within a dataset. Specifically, it introduces SparseMVC, a framework built upon a sparse autoencoder with adaptive constraints, further enhanced by a correlation-informed sample reweighting module designed to alleviate the impact of encoding disparities and constraint-induced inconsistencies.

**Questions:**

- CDA module is designed to align feature distributions across views. Could you provide a more detailed technical explanation of how this alignment is accomplished? Are there any potential risks of overfitting when this alignment is applied too aggressively across views?

- Could the correlation computation between global and view-specific features in the CSR module be compromised by noise introduced from highly sparse views? How is the reliability of the resulting attention weights ensured under such conditions?

- Could you provide a more detailed explanation for the choice of Sv in Eq. (3)?

**Ethical Concerns:**

["NO or VERY MINOR ethics concerns only"]

**Final Justification:**

Thank the author for their rebuttal. All my concerns have been resolved, and I will maintain the score.

**Limitations:**

Yes.

**Paper Formatting Concerns:**

After thoroughly reviewing the paper, no major formatting issues are evident in the provided content. The document appears to adhere to the required guidelines without any noticeable discrepancies.

**Quality:**

4

**Strengths And Weaknesses:**

Strengths:

- Quality: This paper demonstrates a commendable level of quality, with a methodologically sound and well-executed design that reflects careful engineering and a solid understanding of the multi-view clustering problem. The experimental design is rigorous, and the empirical results are convincingly presented, showing clear improvements over existing baselines.
Clarity: In terms of clarity, the paper is written with precision and coherence. The problem is articulated clearly from the outset, and the architectural design is logically motivated and systematically explained. The visualizations are informative and enhance the reader's understanding of the proposed framework.

- Significance: Regarding significance, the paper tackles a meaningful yet underexplored issue in the field—the cross-view sparsity problem—which is both practically relevant and theoretically challenging. By directing attention toward data-level issues rather than solely on architectural refinements, the work offers a compelling shift in perspective that could inspire future research directions.

- Originality: With respect to originality, the proposed SparseMVC framework exhibits an impressive level of conceptual novelty. The integration of adaptive sparsity modeling, sample reweighting based on correlation, and distribution alignment constitutes a multifaceted and coherent strategy that differs from conventional multi-view learning paradigms. Each module contributes distinctly to the overarching objective, and their unified deployment reflects thoughtful innovation.

Weaknesses:

- Although the framework is reasonably efficient, the use of contrastive learning inherently imposes computational overhead, making it unlikely to rank among the fastest available approaches.

- The assumption of well-aligned views may limit its robustness in practical scenarios where slight inter-view misalignments are often unavoidable.

---

> ### Author Rebuttal · Authors · 2025-07-29
>
> Thanks for your acknowledgment of our work. Below, we will address each of your questions.
>
> ---
>
> > ***Question 1**:* **CDA module** is designed to align feature distributions across views. Could you provide a more detailed technical explanation of **how this alignment is accomplished**? Are there any **potential risks of overfitting** when this alignment is applied too aggressively across views?
>
> Thanks. The CDA module aligns feature distributions by drawing together positive pairs between the late-fused global view and each local view and pushing apart negative pairs in the latent space. Technically, this is achieved by minimizing the cosine similarity of diagonal elements while maximizing that of the off-diagonal ones. Based on the convergence analysis , as shown in **Table 6**, our method effectively mitigates the risk of overfitting caused by excessively aligned feature distributions. Notably, the evaluation metrics remain relatively stable, even though the loss fluctuates on small datasets, as the number of alignment iterations increases.
>
> ---
>
> > ***Question 2**:* Could the correlation computation between global and view-specific features in the CSR module be compromised by noise introduced from highly sparse views? How is the reliability of the resulting attention weights ensured under such conditions?
>
> Thanks. First, as shown in **Figure 8** provided in **Appendix B.1**, a 64-dimensional parameter space is sufficient to effectively encode high-dimensional data views, even when the original dimensions (see **Table 2**) differ by up to two orders of magnitude. Moreover, the sparsity ratio in most views remains below 0.9 (less than one order of magnitude), suggesting that the influence of invalid features is largely suppressed during the sparse encoding process. Second, the correlation computation is used to assign sample-level weights, which helps to counteract the impact of sample-wise dimensional collapse. Finally, the attention weights in CSR are computed independently of other modules, ensuring that encoding and alignment constraints do not interfere with correlation estimation.
>
> ---
>
> > ***Question 3**:* Could you provide a more detailed explanation for **the choice of $s_v$** in Eq. (3)?
>
> Thanks. As shown in **Table 1**, we have evaluated several datasets with extreme sparsity $s_v$ variations, such as ALOI-100, where the low sparsity $s_v$ view is 0.0001 and the high sparsity $s_v$ view is 0.6644. Our method includes dynamic autoencoders that adjust encoding and loss constraints based on view sparsity. In datasets with minimal sparsity differences, such as BRCA and Synthetic3d, performance remained at SOTA levels, showing no noticeable degradation. Subsequently, to evaluate whether to select the external scaling factor $f(s_v)$ or the internal ratio coefficient $\rho$, we conducted experiments by keeping one constant and varying the other. The results provided in **Appendix B.2** indicate that optimizing the external scaling factor, while maintaining a fixed internal ratio coefficient, yields the best performance.

---

> > ### Comment · Reviewer_bBMF · 2025-08-05
> >
> > Thank the author for their rebuttal. All my concerns have been resolved, and I will maintain the score.

---

> > > ### Author Response · Authors · 2025-08-05
> > >
> > > Dear Reviewer bBMF,
> > >
> > > We appreciate your attentive feedback. Your suggestions have been invaluable in improving our manuscript, and we are thankful for your ongoing support throughout this process.
> > >
> > > Should you have any further suggestions for enhancement, we would be pleased to receive them. Thank you again for your time and careful consideration.
> > >
> > > Best wishes,
> > >
> > > Authors

---

### Official Review · Reviewer_ar36 · 2025-06-25

**Clarity:** 2
**Significance:** 2
**Originality:** 2
**Rating:** 4
**Confidence:** 4

**Summary:**

This paper proposes a multi-view clustering method focusing on the sparsity difference across views. Thanks to the design of the sparse autoencoder, the correlation-informed sample reweighting module, and the cross-view distribution alignment module, the proposed method achieves the state-of-the-art performance on eight datasets.

**Questions:**

I would like the authors to address my concerns raised in the weaknesses section.

**Ethical Concerns:**

["NO or VERY MINOR ethics concerns only"]

**Final Justification:**

Thanks for the responses. With the authors' explanations and clarifications, my concerns have been addressed, so I would like to raise my score.

**Limitations:**

Yes.

**Quality:**

2

**Strengths And Weaknesses:**

- Strengths
	1. The proposed method is evaluated on eight datasets and achieves promising clustering performance.
- Weaknesses
	1. The motivation of this work is questionable. Why must sparsity correspond to less informative features? The information about zeros depends on its physical meaning behind. For example, true zero values also represent the characteristics of the sample.
	2. While this work claims that sparsity harms the multi-view clustering performance, the method seems to be designed for handling low-quality features in general. In other words, only a few components in the proposed method are specifically designed to handle data sparsity. In fact, the most related design, sparse auto-encoder, has already been proposed in previous works. As a result, the contribution of this work seems limited.
	3. The datasets used for evaluations are too small. Methods evaluated and tuned on datasets with only hundreds of points might not generalize well to other datasets and scenarios.
	4. The proposed method is also sensitive to hyperparameters, which are not easy to tune in practice, given the unsupervised nature of clustering.

---

> ### Author Rebuttal · Authors · 2025-07-30
>
> We sincerely appreciate your constructive comments and suggestions. Below, we will address each of your questions and concerns.
>
> ---
>
> > ***Question 1**:* The motivation of this work is questionable. Why must sparsity correspond to less informative features? The information about zeros depends on its physical meaning behind. For example, true zero values also represent the characteristics of the sample.
>
> **(1) Why must sparsity correspond to less informative features?** Thanks. According to sparse coding and compressed sensing theory [1-2], sparse features lack sufficient capacity to preserve underlying structure and thus carry less informative for downstream tasks. While individual zeros may sometimes be meaningful, our concern lies in the overall representational capacity of a view. For example, in a 1024-dimensional embedding, if only a few dimensions are activated, the representation is likely information-poor. This phenomenon aligns with the theoretical insight that excessively sparse or under‑activated representations—especially when unevenly distributed across views—struggle to capture the full semantics of the data.
>
> **(2) We focus on statistically meaningful patterns of structural sparsity, rather than simplistically equating zero values with a lack of semantic significance.** In our work, sparsity refers to the structural variation across views within the same multi‑view dataset, where some encoded views exhibit extremely limited activation and markedly low information density  [2]. This form of structural sparsity, which frequently produces high-dimensional but low-rank features, limits expressive capacity and complicates alignment and clustering. Therefore, while individual zeros can be meaningful, our study emphasizes statistically significant sparsity patterns across views.
>
> **(3) In multi-omics data, zeros often arise from sequencing dropout or detection limits rather than representing meaningful biological signals [3–4]**. For instance, methylomic and proteomic profiles frequently contain such zeros, while transcriptomic data are typically more complete. In these cases, a zero simply indicates that no signal was detected in that dimension.
>
> **(4) Motivation: addressing view-wise imbalance caused by cross-view sparsity variations, rather than sparsity itself or data‑level sparsity.** In summary, we neither assume that zeros are inherently uninformative nor disregard them. Instead, our framework targets cross-view sparsity variations within the same dataset, an objective statistical property that does not depend on the semantic meaning of zeros. Consequently, we adopt an adaptive and view‑specific encoding strategy tailored to view sparsity differences.
>
> ---
>
> > ***Question 2**:* While this work claims that sparsity harms the multi-view clustering performance, the method seems to be designed for handling low-quality features in general. In other words, only a few components in the proposed method are specifically designed to handle data sparsity. In fact, the most related design, sparse auto-encoder, has already been proposed in previous works. As a result, the contribution of this work seems limited.
>
> We sincerely thank the reviewer for raising this important and insightful concern.
>
> (1) To begin with, we would like to clarify that the primary focus of our work is **not on general data sparsity** or low-quality features, but rather on a more specific and previously underexplored issue: **the sparsity variations across views within the same multi-view data**.
>
> (2) Moreover, our entire framework is specifically designed to handle view-level sparsity variations, a prevalent yet underexplored characteristic of multi-view data, through a complete and tightly integrated data-driven architecture, where **the sparse autoencoder serves only as one means of enforcing sparse encoding**. To address the sparsity variations across views, **we design an adaptive encoding mechanism that leverages the sparsity ratio of each view as prior knowledge, allowing the encoder to transition between standard and sparse modes with appropriate constraint strengths**. To mitigate the enlarged representation gap caused by heterogeneous encoding, we further introduce a correlation-guided reweighting module. It adjusts each view’s contribution during late fusion based on **correlations between late-fused local and early-fused global features**. In addition, a distribution alignment module reinforces inter-view complementarity through structural constraints.
>
> (3) To the best of our knowledge, we are **the first to systematically conduct statistical analysis of view-wise sparsity ratios and formally define the problem of cross-view sparsity variation in multi-view clustering field**. Our contribution lies not only in architectural innovation, but also in the discovery and analysis of fundamental multi-view data characteristics.
>
> (4) Finally, in further ablation summarized in the bottom table, we evaluate two variants: **(i)** all views use sparse autoencoders, and **(ii)** adaptive encoding is applied, both with the CSR reweighting module considered. Compared with above settings, our view-adaptive encoder, which adjusts the encoding strategy based on the sparsity level of each view, consistently achieves superior performance. Results further confirm that the effectiveness arises from the synergy between adaptive encoding and sample reweighting, rather than from the use of sparse autoencoders alone.
>
> |   Method \ Dataset    |           | ALOI-100  |           |           | Dermatology |           |           |  MSRCV1   |           |
> | :-------------------: | :-------: | :-------: | :-------: | :-------: | :---------: | :-------: | :-------: | :-------: | :-------: |
> |                       |    ACC    |    NMI    |    PUR    |    ACC    |     NMI     |    PUR    |    ACC    |    NMI    |    PUR    |
> | all sparse ae w/o CSR |   78.56   |   88.92   |   81.12   |   77.37   |    74.38    |   86.03   |   91.90   |   88.56   |   91.90   |
> |     all sparse ae     |   80.42   |   89.19   |   82.44   |   89.11   |    78.37    |   89.11   |   92.38   |   88.61   |   92.38   |
> |  adaptive ae w/o CSR  |   81.04   |   89.58   |   83.17   |   88.83   |    78.23    |   88.83   |   95.71   |   92.69   |   95.71   |
> |  adaptive ae (ours)   | **82.21** | **92.65** | **84.19** | **95.25** |  **89.86**  | **95.25** | **97.14** | **94.22** | **97.14** |
>
> We hope this work inspires greater attention to the intrinsic characteristics of data and to the design of architectures driven by data.
>
> ---
>
> > ***Question 3**:* The datasets used for evaluations are too small. Methods evaluated and tuned on datasets with only hundreds of points might not generalize well to other datasets and scenarios.
>
> Thanks for the suggestion. We conduct additional experiments on the large-scale datasets: GSE and Animal (please refer to the reference cited for Reviewer p2jA above). Below are the larger dataset details and comparison results using the latest methods employed in our manuscript:
>
> | **Model \ Dataset** |    GSE    | Samples 8,200 | View Sparsity Ratio [0.877, 0.005] |  Animal   | Samples 11,673 | View Sparsity Ratio [0.589, 0.179, 0.355, 0.467] |
> | :-----------------: | :-------: | :-----------: | :--------------------------------: | :-------: | :------------: | :----------------------------------------------: |
> |                     |    ACC    |      NMI      |                PUR                 |    ACC    |      NMI       |                       PUR                        |
> |   MVCAN [CVPR’24]   |   71.40   |     75.41     |               72.29                |   12.06   |     10.25      |                      15.48                       |
> |   SCMVC [TMM’24]    |   62.80   |     74.13     |               65.72                |   16.96   |     15.05      |                      20.15                       |
> |  SparseMVC (ours)   | **73.18** |   **78.42**   |             **74.26**              | **19.90** |   **17.92**    |                    **24.13**                     |
>
> ---
>
> > ***Question 4**:* The proposed method is also sensitive to hyperparameters, which are not easy to tune in practice, given the unsupervised nature of clustering.
>
> Thanks. To begin with, a noticeable performance drop occurs only when the *temperature coefficient $\tau$* ≤ 0.4 or ≥ 1.6. The extremely small value of 0.1 is chosen to probe the lower bound of performance degradation and is rarely used in practical applications. Furthermore, the performance fluctuation mainly occurs along the $\tau$ direction, whereas it remains relatively insensitive to changes in the constraint ratio coefficient $\lambda_{\text{CR}}$. In practice, it is the relative weighting between loss terms that is more commonly adjusted. In addition, the y‑axis was intentionally truncated to better observe the differences, which in turn accentuates the visual disparity. Finally, regarding the concern that hyperparameters are not easy to tune in practice, our method maintains stable performance even under noticeable loss fluctuations, as illustrated in Fig. 6.
>
> **Reference**
>
> [1] Donoho D L. Compressed sensing[J]. IEEE Transactions on information theory, 2006, 52(4): 1289-1306.
>
> [2] Donoho D L, Elad M. Optimally sparse representation in general (nonorthogonal) dictionaries via ℓ1 minimization[J]. Proceedings of the National Academy of Sciences, 2003, 100(5): 2197-2202.
>
> [3] Cao Z J, Gao G. Multi-omics single-cell data integration and regulatory inference with graph-linked embedding[J]. Nature Biotechnology, 2022, 40(10): 1458-1466.
>
> [4] Baysoy A, Bai Z, Satija R, et al. The technological landscape and applications of single-cell multi-omics[J]. Nature Reviews Molecular Cell Biology, 2023, 24(10): 695-713.

---

> > ### Comment · Reviewer_ar36 · 2025-08-04
> >
> > Thanks for the explanations and clarifications. My concerns have been addressed, and I will raise my score accordingly.

---

> > > ### Author Response · Authors · 2025-08-04
> > >
> > > Dear Reviewer ar36,
> > >
> > > We are truly thankful for your reply. Your valuable feedback has significantly improved our manuscript, and we are grateful for your continued support throughout this process.
> > >
> > > If there are any further aspects you believe could benefit from refinement, please feel free to share your thoughts. Thank you again for your time and thoughtful consideration.
> > >
> > > Best wishes,
> > >
> > > Authors

---

### Official Review · Reviewer_p2jA · 2025-06-27

**Clarity:** 3
**Significance:** 3
**Originality:** 3
**Rating:** 5
**Confidence:** 4

**Summary:**

This paper addresses the varying sparsity problem in multi-view data by proposing an adaptive sparse autoencoder for MVC. To mitigate encoding disparities across views, an attention mechanism is incorporated based on inter-view correlations. Furthermore, to enhance representation consistency, a contrastive loss is introduced to align view-specific representations with a unified global representation.

**Questions:**

1. How is multi-view data fused in the early fusion strategy?
2. Why does the sparsity of the ALOI-100 dataset differ between Figure 1 and Table 1? There appears to be an inconsistency in the reported sparsity of the ALOI-100 dataset between Figure 1 and Table 1.

**Ethical Concerns:**

["NO or VERY MINOR ethics concerns only"]

**Final Justification:**

After carefully reading the rebuttal, I found that most of my concerns have been fixed, including adding more large-scale datasets which also exhibit SOTA performances, and more experiments w.r.t the effectiveness of the SAA module. So I would like to raise the score to acceptance.

**Limitations:**

YES

**Quality:**

3

**Strengths And Weaknesses:**

Strength:
This paper tackles cross-view sparsity variation in multi-view learning and conducts experiments on datasets with pronounced sparsity differences across views, such as ALOI-100.
Weakness:
1.	The method addresses sparsity variation by applying an adaptive weight based on an existing entropy-matching loss. This solution is relatively simplistic; additional theoretical justification is recommended to support the claimed contribution.
2.	Most of the evaluated datasets are small-scale, typically containing only a few hundred samples, which may limit the robustness of the conclusions.
3.	The method lacks comparisons with MVC approaches specifically designed to handle data-level sparsity. An ablation study of the adaptive weight component is also necessary to validate the effectiveness of the proposed SSA. As currently presented, the results do not convincingly argue the method’s ability to handle varying sparsity.

---

> ### Author Rebuttal · Authors · 2025-07-29
>
> Thanks for your constructive reviews and valuable suggestions. Below, we will address your questions and concerns one by one.
>
> ---
>
> > ***Question 1**:* How is multi-view data fused in the early fusion strategy?
>
> Thanks. The early fusion strategy integrates the data from each view through **feature concatenation**. Since our method uses the global latent representation encoded from early fusion to guide the subsequent dynamically weighted late fusion of local representations, we chose an early fusion approach that preserves the original feature values as much as possible.
>
> ---
>
> > ***Question 2**:* Why does the sparsity of the ALOI-100 dataset differ between Figure 1 and Table 1? There appears to be an inconsistency in the reported sparsity of the ALOI-100 dataset between Figure 1 and Table 1.
>
> Thanks, and I apologize for any misunderstanding or confusion. As explained in the figure caption, the sparsity ratios in **Figure 1** are transformed using **the sigmoid function**, which shifts the baseline from 0 (the bottom of the image) to 0.5 (the middle of the image) for better visualization, as many views have very small sparsity values (less than 0.01) that would be hard to see with the original scale.
>
> ---
>
> > ***Question 3**:* The method addresses sparsity variation by applying an adaptive weight based on an existing entropy-matching loss. This solution is relatively simplistic; additional theoretical justification is recommended to support the claimed contribution.
>
> Thanks. The entropy-matching loss serves merely as one component for regulating the sparsity of representations, and our contribution is by no means limited to this design alone. Rather, by systematically defining, quantifying, and analyzing cross-view sparsity variation as a fundamental characteristic of multi-view data, we construct an integrated, data-driven framework that is specifically motivated by this widely existing yet previously underexplored property. To address sparsity variations across views within a dataset, we propose not only an adaptive encoding strategy that enables the encoder to automatically transition between standard and varying degrees of sparse encoding, but also a series of interdependent mechanisms to mitigate the side effects of representational divergence caused by such differentiated encoding. Specifically, we incorporate a correlation-guided fusion strategy that leverages global-to-local feature relationships established in the early stage to guide the weighting of local features during late fusion. Moreover, a distribution alignment module further constrains the fused representations structurally, thereby enhancing cross-view complementarity in the final stage.
>
> Regarding the theoretical analysis, we have discussed the necessity of adaptive sparsity constraints in a summarized manner in Appendix E, where we explain how uniform encoder architectures and fixed loss functions are often ineffective in handling sparsity variations across different views. We also outline how our method balances compression and expressiveness by adjusting the strength of sparsity constraints, which in turn enhances the alignment of latent representations. We hope this partially addresses your concerns, and, in line with your suggestion, we will further expand this section to more comprehensively substantiate the contributions of our method.
>
> ---
>
> > ***Question 4**:* Most of the evaluated datasets are small-scale, typically containing only a few hundred samples, which may limit the robustness of the conclusions.
>
> Thanks for the suggestion. We conduct further experiments on the large-scale datasets GSE [1] and Animal [2]. Below are the larger dataset details and comparison results using the latest methods employed in our manuscript:
>
> | **Model \ Dataset** |    GSE    | Samples 8,200 | View Sparsity Ratio [0.877, 0.005] |  Animal   | Samples 11,673 | View Sparsity Ratio [0.589, 0.179, 0.355, 0.467] |
> | :-----------------: | :-------: | :-----------: | :--------------------------------: | :-------: | :------------: | :----------------------------------------------: |
> |                     |    ACC    |      NMI      |                PUR                 |    ACC    |      NMI       |                       PUR                        |
> |   MVCAN [CVPR’24]   |   71.40   |     75.41     |               72.29                |   12.06   |     10.25      |                      15.48                       |
> |   SCMVC [TMM’24]    |   62.80   |     74.13     |               65.72                |   16.96   |     15.05      |                      20.15                       |
> |  SparseMVC (ours)   | **73.18** |   **78.42**   |             **74.26**              | **19.90** |   **17.92**    |                    **24.13**                     |
>
> ---
>
> > ***Question 5**:* The method lacks comparisons with MVC approaches specifically designed to handle data-level sparsity. An ablation study of the adaptive weight component is also necessary to validate the effectiveness of the proposed SSA. As currently presented, the results do not convincingly argue the method’s ability to handle varying sparsity.
>
> Thanks. Our approach focuses on view-level structural sparsity, specifically the sparsity variation across views within the same multi-view data. This differs from data-level sparsity methods, which typically apply uniform sparse encoding to all views without explicitly considering the heterogeneity of inter-view sparsity. To further **validate the effectiveness of the proposed SAA module**, we extend the ablation study in Table 5 by introducing two additional comparative settings: **(i) *uniformly sparse encoding applied to all views, which mimics methods designed for data-level sparsity***; and **(ii)** ***adaptive encoding tailored to each view*.** On top of this, **we also ablate the CSR module**, which reweights the local features during the late fusion stage. Regardless of whether the CSR module is applied, the results in the bottom table show that the proposed SAA, which leverages adaptive autoencoders, remains effective and consistently achieves superior performance.  Collectively, these findings confirm that our method is robust to varying sparsity across views and that the SAA and CSR modules function synergistically rather than independently.
>
> |   Method \ Dataset    |           | ALOI-100  |           |           | Dermatology |           |           |  MSRCV1   |           |
> | :-------------------: | :-------: | :-------: | :-------: | :-------: | :---------: | :-------: | :-------: | :-------: | :-------: |
> |                       |    ACC    |    NMI    |    PUR    |    ACC    |     NMI     |    PUR    |    ACC    |    NMI    |    PUR    |
> | all sparse ae w/o CSR |   78.56   |   88.92   |   81.12   |   77.37   |    74.38    |   86.03   |   91.90   |   88.56   |   91.90   |
> |     all sparse ae     |   80.42   |   89.19   |   82.44   |   89.11   |    78.37    |   89.11   |   92.38   |   88.61   |   92.38   |
> |  adaptive ae w/o CSR  |   81.04   |   89.58   |   83.17   |   88.83   |    78.23    |   88.83   |   95.71   |   92.69   |   95.71   |
> |  adaptive ae (ours)   | **82.21** | **92.65** | **84.19** | **95.25** |  **89.86**  | **95.25** | **97.14** | **94.22** | **97.14** |
>
> **Reference**
>
> [1] Danford T, Rolfe A, Gifford D. GSE: a comprehensive database system for the representation, retrieval, and analysis of microarray data[C]//Pacific Symposium on Biocomputing. Pacific Symposium on Biocomputing. 2008: 539.
>
> [2] Lampert C H, Nickisch H, Harmeling S. Attribute-based classification for zero-shot visual object categorization[J]. IEEE transactions on pattern analysis and machine intelligence, 2013, 36(3): 453-465.

---

> > ### Comment · Reviewer_p2jA · 2025-08-03
> >
> > Thanks for the author's rebuttal. After carefully reading the rebuttal, I am happy to see that most of my concerns have been fixed, including adding more large-scale datasets which also exhibit SOTA performances, and more experiments w.r.t the effectiveness of the SAA module. So I would like to raise the score in the final voting stage.

---

> > > ### Author Response · Authors · 2025-08-04
> > >
> > > Dear Reviewer p2jA,
> > >
> > > We sincerely appreciate your feedback. Your constructive comments have been instrumental in enhancing our work, and we are grateful for the time and attention you have dedicated to it.
> > >
> > > If you have any further suggestions for improvement, please feel free to share them. Thank you again for your time and thoughtful consideration.
> > >
> > > Best wishes,
> > >
> > > Authors

---

### Note · Authors · 2025-08-13

We are truly grateful for the helpful suggestions and detailed feedback provided by all the reviewers.

We also sincerely appreciate the Area Chair's valuable time, dedicated effort, and kind attention throughout the entire review process.

During the rebuttal phase, **we provided detailed and point-by-point responses to all reviewers’ comments**, including additional comparative experiments on large-scale multi-view datasets and a more thorough ablation study of the SAA module, which further demonstrate the innovation and robustness of SparseMVC. We also clarified misunderstandings regarding the sparsity ratio in Figure 1 and hyperparameter sensitivity. Moreover, we addressed concerns related to sparsity versus zero values, the compression layer, and the sparse threshold. All reviewers gave positive feedback, and no new issues were raised. We hope these responses have adequately addressed any remaining concerns from the reviewers, Area Chairs, and Program Chairs.

**It is encouraging to see that Reviewers h93U, bBMF, ar36, and p2jA have recognized the key strengths and significant contributions of our work**, such as:

- an effective multi-view clustering method for addressing cross-view sparsity variation, supported by extensive experimental comparisons ( Reviewer p2jA );
- clear motivation, focus on a meaningful yet underexplored issue, rigorous experimental design, and the potential to inspire future research from a new perspective ( Reviewer bBMF );
- comprehensive experiments across benchmarks and promising clustering performance ( Reviewer ar36 );
- a refined and effective solution that addresses a challenge overlooked in largely prior research, along with exceptional clarity and a well-structured presentation ( Reviewer h93U ).

Thanks & Best regards,

 **Authors of paper-8849**

---

### Decision · Program_Chairs · 2025-09-17

**Decision:**

Accept (spotlight)

**Comment:**

The paper studies an underexplored problem (i.e., cross-view sparsity variation) and proposes SparseMVC, a coherent three-part solution (adaptive sparse autoencoder, correlation-informed sample reweighting, and cross-view distribution alignment). Reviewers found the problem meaningful, the design well-motivated, and the empirical results convincing (especially after the authors’ rebuttal added larger-scale experiments (GSE, Animal) and stronger ablations), leading to a consensus recommendation to accept. Remaining concerns to address in the final version include stronger theoretical justification for the entropy-matching/adaptive weighting choices, clearer delineation of novelty versus reused components, more practical guidance on hyperparameter sensitivity and computational costs, and improved figure/notation clarity; the authors should also release code and reproduction scripts. Overall, the work could advance multi-view clustering by reframing a data-driven challenge and delivering an effective, empirically validated approach.